# Combination of Biochemical, Molecular, and Synchrotron-Radiation-Based Techniques to Study the Effects of Silicon in Tomato (*Solanum Lycopersicum* L.)

**DOI:** 10.3390/ijms232415837

**Published:** 2022-12-13

**Authors:** Marta Marmiroli, Francesca Mussi, Valentina Gallo, Alessandra Gianoncelli, William Hartley, Nelson Marmiroli

**Affiliations:** 1Department of Chemistry, Life Science and Environmental Sustainability, University of Parma, Parco Area delle Scienze 33/A, 43124 Parma, Italy; 2The Italian National Interuniversity Consortium for Environmental Sciences (CINSA), Parco Area delle Scienze 93/A, 43124 Parma, Italy; 3Elettra-Sincrotrone Trieste, Strada Statale 14—km 163.5 in AREA Science Park, Basovizza, 34149 Trieste, Italy; 4Agriculture and Environment, Harper Adams University, Newport B5062, UK

**Keywords:** tomato, silicon, elemental imaging, physiological analyses, chemical analyses, molecular analyses

## Abstract

The work focused on the analysis of two cultivars of tomato (*Solanum lycopersicum* L.), Aragon and Gladis, under two different treatments of silicon, Low, 2 L of 0.1 mM CaSiO_3,_ and High, 0.5 mM CaSiO_3_, weekly, for 8 weeks, under stress-free conditions. We subsequently analyzed the morphology, chemical composition, and elemental distribution using synchrotron-based µ-XRF techniques, physiological, and molecular aspects of the response of the two cultivars. The scope of the study was to highlight any significant response of the plants to the Si treatments, in comparison with any response to Si of plants under stress. The results demonstrated that the response was mainly cultivar-dependent, also at the level of mitochondrial-dependent oxidative stress, and that it did not differ from the two conditions of treatments. With Si deposited mainly in the cell walls of the cells of fruits, leaves, and roots, the treatments did not elicit many significant changes from the point of view of the total elemental content, the physiological parameters that measured the oxidative stress, and the transcriptomic analyses focalized on genes related to the response to Si. We observed a priming effect of the treatment on the most responsive cultivar, Aragon, in respect to future stress, while in Gladis the Si treatment did not significantly change the measured parameters.

## 1. Introduction

Tomato (*Solanum lycopersicum* L.) is one of the most consumed fruits in the world [1], and its popularity is increasing constantly. The high consumption of fresh, cooked, or processed tomatoes makes it one of the principal sources of vitamins and minerals in the human diet [2]. Today Asia dominates the tomato market, with China ranking first, followed in decreasing order by India and the USA [3]. *Solanum* belongs to the economically important plant family of Solanaceae and plays a pivotal role as a model plant species for fleshy fruits because of its agronomic and genetic features, and particularly as a rich plant source of carotenoids, vitamins, and minerals [4,5]. The tomato has many interesting features that other model plants, such as rice and *Arabidopsis*, do not have. For example, besides bearing fleshy fruits, tomato plants have sympodial shoots and a mostly indeterminate (vine) growth habit (with some exceptions, see the Micro-Tom cultivar). Most of these traits are agronomically important and have been the major targets for domestication and breeding [6]. Improving nutritional quality by enhancing the contents of bioactive compounds has become an important aspect for tomato fruit quality valorization [7]. The tomato fruit is a model system for fleshy fruit ripening, making it an ideal system to assess genes, processes, and environmental conditions affecting fruit ripening and quality [8]. Fruit ripening involves the acquisition of a series of fruit traits that make the fruit attractive and palatable [9]. The conversion from chloroplast to chromoplast is an important part of the ripening process and is normally associated with the dismantling of the photosynthetic machinery and the accumulation of carotenoids in chromoplasts and sugars, organic acids, and volatile aromatic compounds in the fruit cells [8,10]. The entire genome of *S. lycopersicum* (the inbred cultivar “Heinz 1706”) was sequenced in 2012 [11], which allowed for the development of a variety of bioinformatic tools for genomics/transcriptomics, proteomics, and metabolomics [12]. Silicon (Si) is the second most abundant element in the earth’s crust after oxygen [13], and it is present principally as SiO_2_ [14]. It is nearly ubiquitously available to plants that absorb Si from the soil only in the form of silicic acid (Si(OH)_4_) [15]. Soluble monosilicic acid H_4_SiO_4_ normally ranges from 0.1 to 0.6 mM in the soil solution [16], although the exact availability varies depending on soil type, clay or sandy, pH, and temperature [14]. In soil solution, Si is present as silicic acid, Si(OH)_4_, at pH < 9, at concentrations between 0.1 mM and 2.0 mM, which is in the same order of magnitude as potassium, calcium, and other major plant nutrients [17,18].

A part of the total Si is taken up passively [19,20,21], but especially in Si accumulators, Si is mainly taken up actively through specific transporters [22,23,24,25]. In Si accumulators, Lsi1 and Lsi2, two Si transporters are necessary to transport silicic acid from soil through the root tissues and the casparian strip [23,26]. To unload silicic acid from the xylem to the shoot, Lsi6, a homolog of Lsi1, is necessary [27,28,29]. Lsi1 has been characterized in several species such as wheat [30], rice [23], maize [28], barley [31], soybean (*Glycine max* L.) [32], pumpkin (*Cucurbita moscata Dutch*.) [33], and cucumber (*Cucumis sativus*) [34]. Lsi1, providing primary entry of Si(OH)_4_ into plant root cells (and, to a lesser extent, arsenious acid (As(OH)_3_) and boric acid (B (OH)_3_) [35], belongs to the superfamily of major intrinsic proteins (MIPs, also known as aquaporins (AQPs)) [23]. Lsi2 is a member of an anion transporter family. It has eleven predicted transmembrane domains and is localized to the plasma membrane [26]. Regarding Lsi2, the fundamental issue of the mechanism of transport remains obscure. It is thought that Lsi2 belongs to a class of putative anion transporters, showing similarity with the arsenite efflux transporter ArsB from bacteria and Archaea, and functions as a Si(OH)_4_/H^+^ antiporter [26]. However, direct evidence for this transport process is currently lacking [36].

Mechanisms of Si adhesion and accumulation are also opaque. Increase attention has been dedicated recently to this topic [20,21,37]. Si is mostly found polymerized in the apoplast (e.g., around exodermal and endodermal root cells and leaf epidermal cells) [38,39], and cell wall constituents, such as (hemi)cellulose, callose, pectin, and lignin, have been demonstrated to interact with Si(OH)_4_ as “scaffolding” or “templates” for silicification [21,36]. Deposited silica can be found as phytoliths occurring in a multitude of shoot tissues [40]. Silica can also accumulate in or beneath the cuticle layer of the cell wall in epidermal cell layers and tissues surrounding the vasculature [20,37,41]. Although not considered an essential element in the canonical sense [42], it is now considered a “beneficial substance” or “quasi-essential” [43]. The reason for this promotion is that Si can help plants to resist all possible biotic and abiotic stresses [44]. Notwithstanding some effects of Si on plant metabolism and gene expression that have been demonstrated, the mechanisms of action of Si on plants and its capacity to alleviate stresses remain unclear [45].

Many scientific investigations have been conducted on the capacity of Si to alleviate abiotic stress in tomato (see Appendix A) by influencing its metabolism and transcription, but none have been conducted on the effect of Si on plants that were not under the effects of stress.

In this study, we aimed to indagate the effects of two different concentrations of Si (low and high) on the physiological and molecular response of two cultivars of tomato, contrasting in their ability to uptake As [46]. The rationale of the experiment was to highlight possible responses in the two tomato cultivars when under the treatment with Si during their life cycle, in normal conditions. We analyzed markers of oxidative stress, enzymatic and non-enzymatic, the concentrations in macro and micronutrients, including the use of the synchrotron-based technique µ-XRF to map the distribution of Si within fruits, leaves and roots, and the level of expression of genes related to mitochondrial, chloroplastic, and stress-related functions, to identify possible interactions with Si and plant metabolism and cells under normal conditions. Thus far, all the papers on Silicon have been on its effects of stress mitigator, especially in tomato.

## 2. Results

### 2.1. Morphological Analysis

There were no significant differences among the treatments and the cultivars for any of the morphological traits we examined: aerial parts, length and weight, roots, length and weight, and fruit number and weight. However there seemed to be a trend within the two cultivars. In cv Aragon, all the values of the traits tended to diminish with an increase in administered Si, while in cv. Gladis there seemed to be less difference between the control and the treatments for all the traits, but only at the level of trends (Appendix A). The means for the aerial parts are almost the same, but with higher standard errors in respect to the roots and the fruit. If we proceed with a Canonical Discriminant Analysis utilizing the traits as dependent variables and the Cultivars and the Treatments as independent variables, it appeared clear that the effect of the Si treatment on the morphological traits is cultivar-dependent (Appendix A). The Canonical Discriminant Function Analysis utilizes the two Canonical Functions that result from the Factor reduction of the groups of independent variables (three in our case: Control, Si Low, Si High) to create a plot where the first canonical Function is on the x axis and tells us how much variability exists among groups of dependent variables and the second Canonical function is on the y axis and tells us how much variability exists within each group of independent functions.

The behavior of tomato plants towards the Si treatments was clearly cultivar-dependent for the physiological and biochemical analyses and for the chemical concentrations of the main elements analyzed, as can be visualized from the Discriminant Canonical Analysis (Figure 1, Figure 2, and Appendix A). Therefore, we decided to treat the two cultivars singularly in the results section and to limit the comparison between cultivars in the discussion section.

### 2.2. Physiological and Biochemical Analyses

#### 2.2.1. Cultivar Aragon: Si Concentration in the Plant Parts

The Low Si treatment did not cause differences in Si concentrations in the aerial parts of the plants from the control, but in the root, it appeared that the concentration was lower in treated plants. With the High Si treatment, the concentration of Si in the roots was higher than the control, while in the leaf and fruit it was comparable with the control one and with the Low Si treatment (Appendix A).

#### 2.2.2. Cultivar Aragon: Oxidative State of the Plant Parts

The concentration of H_2_O_2_ in the leaf increased significantly with the two treatments in respect to the control, more with the Low Si treatment, while the concentration of H_2_O_2_ diminished in the roots and fruits, again more with the Low Si treatment. The lipid peroxidation tended to increase in all the three parts of the plant with increasing Si, but the trend did not differ from the control. The antioxidants Total Phenolics (TP) were high in the leaves in general and lower in roots and fruits and the treatments with Si did not change this trend. The treatment with Low Si increased the content of TP in leaves significantly, more than the treatment with High Si. Proline concentration did not change in the root from the control to the two treatments. The Low Si treatment proline in leaf and fruit decreased in comparison to the control, while in the treatment with High Si, proline increased in both leaf and fruit (Figure 3a,e).

#### 2.2.3. Cultivar Aragon: ROS (Reactive Oxygen Species) Enzymes Scavengers

The ascorbate peroxidase (APX), which is a H_2_O_2_ and hydroxyl radicals (^•^OH) scavenger, followed the same trend in the control and in the two treatments, being greater in the leaves and lower in roots and fruits. The treatment with High Si lowered APX in leaves and increased it in fruits. The catalase enzyme (CAT), which catalyzes the decomposition of H_2_O_2_ to water and oxygen, diminished in roots and leaves in respect to controls with the treatment with Low Si, while the treatment with High Si increased CAT in roots and leaves. The concentration of CAT in the fruit did not change from the control with both treatments. The Superoxide Dismutase (SOD), an enzyme that alternately catalyzes the dismutation of the superoxide (O^−^_2_) radical into oxygen (O_2_) and hydrogen peroxide (H_2_O_2_), was increased in all plant parts by the Low Si treatment in comparison to controls and was diminished with the High Si treatment (Figure 3b,d–f). Glutathione peroxidase (GPx) is the general name of an enzyme family with peroxidase activity, and its biochemical function is to reduce lipid peroxides to their corresponding alcohols and to reduce free hydrogen peroxide to water. In general, this enzyme was higher in roots and fruits and lower in leaves. The two treatments did not change this trend. The High Si treatment increased GPx, especially in the fruit. Glutathione S-Transferase (GST), which is a family of enzymes that catalyzes the conjugation of the reduced form of glutathione (GSH) to xenobiotic substrates for the purpose of detoxification, in the control is very high in the leaf and low in roots and fruit. The two treatments decrease the value of GST in leaves and increase it in fruit in comparison with the control. Lipoxygenase (LOX) catalyzes the oxidation of polyunsaturated fatty acid into fatty acid hydroperoxides. In the control it was high in the fruit and lower in root and leaves. The treatment with Low Si diminished the enzyme in the three plant parts, but kept the same trend, while the treatment with High Si, increased LOX in leaves and reduced it in fruits (Figure 3f–k). These parameters are all relevant to this study because they provide the state of the oxidative balance within the cells of the different plant organs which is the scope of Si treatments. In conditions of stress, we would expect these parameters to follow different trends from those described above, where the antioxidant enzymes increase to counter the attack of ROS to the various cell parts.

#### 2.2.4. Cultivar Gladis: Si Concentration in the Plant Parts

The concentration of Si in the cultivar Gladis was higher in the control, especially in the root, in comparison to the two treatments, which did not have any effect on the Si concentrations in the other parts of the plant (Appendix A). Interestingly, this can support the idea that Si might be adhered to the root.

#### 2.2.5. Cultivar Gladis: Oxidative State of the Plant Parts

H_2_O_2_ increased in leaf and root in comparison to the control, under both treatments, but in the Low Si treatment it was higher in the leaves than the roots, while in the High Si treatment the opposite occurred. H_2_O_2_ in the fruit was always very low, and not detected in the treatment with Low Si. Lipid peroxidation in the roots did not change with any of the treatments, while in the leaves it decreased compared to controls with Low Si treatments and increased with the High Si treatments. MDA in the fruit decreased in respect to control under both treatments. Total phenolics were always higher in the leaf in respect to the root and fruit and were diminished by the Low Si treatment. In the root, TP were increased in respect to control in the High Si treatment. TP in the fruit did not change from control to treatments. None of the treatments changed proline in roots, but both treatments diminished proline content in fruit in comparison to the control. Proline in leaves was increased by the High Si treatment and decreased by Low Si treatment in respect to controls (Figure 4a–c and relative Tukey’s tables). Proline usually increased in all the plant parts in condition of osmotic and oxidative stress because it has the double function to promote water retention and turgor and to quench ROS. Here, it is important to notice that the proline trend does not follow the Si concentrations treatments, as it is found usually in a condition of stress [47].

#### 2.2.6. Cultivar Gladis: ROS (Reactive Oxygen Species) Enzymes Scavengers

APX in roots did not change under any of the treatments with respect to controls. In leaves it diminished in the Low Si treatment and increased in the High Si treatment with respect to controls. APX in fruits increased under both treatments. Catalase decreased in roots in both treatments in comparison to controls. CAT in leaves increased in the low treatment when compared to controls, but it decreased in the High Si treatments. CAT in fruit diminished with the High Si treatment both in respect to the Low Si treatment and to controls. SOD in roots and leaves in the Low Si treatment did not change in comparison to controls, but it decreased in both root and leaves. SOD in fruit increased in both treatments. POD in leaves of the control was the highest value, but in the two treatments, POD in leaves diminished substantially. POD in roots did not change in respect to controls in the Low Si treatment but it increased in the High Si treatment. In the fruit, POD decreased in treatments in comparison to the control (Figure 4d,g–k and relative Tukey’s tables). The highest value of GPX was found in the fruit of the Low Si treatment; in the High Si treatment, GPX was higher than the control. GPX in roots was diminished by both treatments in respect to controls. In the leaves, GPX was not affected by the treatments. GST diminished in leaves of treated plants in comparison to controls, but the roots and the fruits were not affected by the treatments. LOX in roots was increased by the two treatments in comparison to controls, however it was decreased in both leaves and fruits by the two treatments in comparison to the controls (Figure 4a–k and relative Tukey’s tables). As we have observed in the cultivar Aragon, there was not a trend followed by the antioxidant enzymes that points to the homogeneous response to a stress, but rather an increase and decrease linked to the plant parts.

### 2.3. Chemical Analyses

All the elements analyzed were affected by the treatments changing from the control situation to the treatment with Low or High Si.

#### 2.3.1. Elements in Cv. Aragon

Appendix A represent the chemical analyses of the elements in the various organs according to treatment and to the plant parts, respectively, for the cultivar Aragon. Different letters are for significantly different values with *p* < 0.005 according to ANOVA and Tukey SHD post hoc test. In Table 1, the two-way ANOVA of the part of the plants, root, shoot, and fruit is reported; the treatments, no treated, Si Low, Si High, and their interaction, with the values of the F statistics, the *p* value, and the asterisks assigned according to the *p* value: *p* ≤ 0.001 = ***; *p* ≤ 0.005 = **; *p* ≤ 0.01 = *; *p* > 0.01 = ns.

This interaction between parts of the plants and treatments was present for Mn, Na, K, P, Mg, and Si. For macronutrients Ca, K, and P, both treatments diminished the concentrations of these elements in the leaves with respect to the control and increased in the roots. The concentration of Ca in the fruit was diminished by the two treatments, while for K and P, the treatments did not change their concentrations in respect to the control (Appendix A). For other nutrients such as Na, Fe, and Mg, the behavior was unique. For Na, both treatments increased the concentration in the leaves and did not change that in roots and fruits in comparison to the controls. For Fe, the Low Si treatment increased the concentrations in all plant parts with respect to the control, while the High Si treatment decreased Fe concentrations in all plant parts (Appendix A). For Mg, the Low Si treatment decreased the concentrations in leaves and fruits, while it increased the concentration in roots; the High Si treatment decreased the concentration of Mg in all plant parts in comparison to the control (Appendix A). Regarding the micronutrients Zn and Al, the Low Si treatments increased the concentrations in all plant parts, while the High Si treatment increased the concentrations even more in all plant parts in comparison to controls. For Al, it greatly diminished all the concentrations. For Cu, the Low Si treatment increased the concentrations in all plant parts, while the High Si treatment diminished the concentrations in leaves and roots and increased that in fruit in comparison to the control. For Mn, both the treatments diminished the concentration in the leaves and left the concentrations in roots and fruits unchanged with respect to controls (Appendix A). For Mo, the Low Si treatment decreased the concentration in roots and fruits and increased that in leaves in comparison to the control; the High Si treatment decreased the concentrations in all plant organs with respect to controls (Appendix A).

#### 2.3.2. Elements in Cv. Gladis

Appendix A represent the chemical analyses of the elements in the various organs according to treatment and for the plant parts, respectively, for the cultivar Gladis. ANOVA and Tukey SHD post hoc test are reported in Appendix A. In Table 1, the two-way ANOVA of the part of the plants, root, shoot, and fruit is reported; the treatments, non-treated, Si Low, Si High, and their interaction, with the values of the F statistics and the *p* value. The interactions between plant parts and treatments are for the following elements: Cu, Mn, Na, Zn, K, and Mg. Considering Ca, K, and P, both the treatments diminished the concentrations in the leaves in respect to control but had different effects on the other plant parts. For Ca, the Low Si treatment diminished the concentrations in the roots and increased that in the fruit, and for the High Si treatments this was the opposite, always in respect to the control values (Appendix A and relative Tukey’s tables). For K, the Low Si treatment diminished the concentrations in leaf and root, but not in the fruit in comparison to the control the High Si treatment produced the same behavior. The concentration in the fruit increased with respect to the control (Appendix A and relative Tukey’s tables). For P, the Low Si treatment increased the concentration in roots and decreased that in leaf and fruits in comparison to the control, while the High Si treatment greatly increased the concentration in the root, leaving the concentrations in roots and fruits unchanged with respect to control (Appendix A and relative Tukey’s tables). Regarding Na, Fe, and Mg, all the treatments diminished the concentrations in the leaf in respect to control, but the trend for the other plant parts was very different. For Fe, both the Si treatments reduced the concentration in the root in respect to controls, without changing the concentration in the fruit. For Na, the Low Si treatment reduced the concentrations in the root, and in the fruit, the High Si treatment reduced the concentrations in the roots and in the shoot with respect to controls, but slightly increased the concentration in the fruit (Appendix A and relative Tukey’s tables). For Mg, the Low Si treatment reduced the concentrations in root and fruit in comparison to treatments, while the High Si treatment reduced the concentration in the root but increased that in the fruit in respect to controls. Considering the micronutrients Zn and Al, both the treatments decreased the concentrations in roots and shoots, especially for Al, but not in the fruit with respect to controls. Additionally, for Cu, Ni, and Mn, both the treatments greatly diminished the concentrations in the roots and shoots, leaving the concentration in the fruit almost unchanged with respect to controls (Appendix A and relative Tukey’s tables). Mo had a very peculiar behavior in the Low Si treatment, where the concentrations greatly increased in shoots and fruits and diminished in the roots in respect to controls, while for the High Si treatment, the concentrations in all the plant parts diminished (Appendix A and relative Tukey’s tables).

### 2.4. µ-XRF (Micro–X-ray Fluorescence) for Elements Distribution

As explained in the Materials and Method section, we utilized this technique to visualize the cellular distribution principally of Si in root, leaf, and fruit, in the control, and under the High Si treatment of the two cultivars Aragon (Figure 5a, Figure 6 and Figure 7) and Gladis (Figure 5b, Figure 8 and Figure 9). Additionally, other elements were visualized: O, Na, Mg, and Al. The behaviors of Si have been inserted because they are also important for the homeostasis of the plants and to understand the mechanisms after Si treatments. Except for Aluminum, their distribution allows for better visualization of the sample morphology, as they are endogenous elements. There are no maps for the treatments Si Low because we did not have enough beamtime at the synchrotron to also perform those analyses. There are no maps for the treatments Si Low because we concentrated on the High treatment, because this was the only one with a chance to translocate Si to the edible parts of the plants and thus have a relevance for food safety.

#### 2.4.1. Si Distribution in Aragon

The distribution of Si within the cell root for the High treatment is mainly on the cell wall. It is distributed in the parenchyma cells (Figure 5a), following the same pattern as Na and Mg. Aluminum is distributed inside the root in a uniform fashion within the cell, although slightly more on the cell wall (Figure 5a).

In fruit control (Figure 6a) there is a marked intense point of Si in a cell, co-localized with a hot spot of Al and with higher content of Mg, Na, and O as well. Therefore, it is likely a Si salt or on an artefact of the preparation. By rescaling the intensity counts of the map, it seems that Si is present quite uniformly inside the fruit. In Aragon fruit under the high treatment (Figure 6b) Si has a clearer pattern, as it appears to be distributed inside the cell with higher concentrations in the center of the pericarp cell, and partially on the cell walls. The other elements are more intensely distributed in the cell walls. In the leaf control (Figure 7a), beside some hot spots, Si is mildly distributed in some cell walls, while in the leaf under the high treatment (Figure 7b), Si is clearly visible with Na, Mg, and Al in the cell walls.

#### 2.4.2. Si Distribution in Gladis

Silicon distribution in Gladis roots under the High treatment (Figure 5b) was mainly in the parenchyma and endodermis tissues, especially in the cell walls. Na and Mn follow the same distribution, even if less concentrated. Aluminum is distributed mainly on the cell walls of the parenchyma and endodermis (Figure 5b). From the maps, Si content in Gladis-treated roots appear to be lower than on Aragon-treated roots (Figure 5a).

In Gladis control fruits (Figure 8a), Si seems to be distributed mainly along cell walls. It is possible to see that the other elements are also distributed in the cell walls of the pericarp. For the fruit treated with High concentrations of Si (Figure 8b), the element penetrated inside the cells, while Mg and Na remained in the cell wall. Al has a high concentration spot, corresponding to a high concentration spot in the Si map, which can correspond to an aluminosilicate aggregate. In the control of the leaf (Figure 9a), Si has a high intensity spot corresponding to the same position in the Al map, which could be an aluminosilicate aggregate. Beside this, it appears to be distributed in the cell walls, co-localizing with Mg and Na. For the map of the High Si treatment (Figure 9b), it is possible to confirm that Si is both within the leaf cells and on the cell walls, while Na, Al, and Mg are mainly on the cell walls.

### 2.5. Molecular Analyses

The RNA for the transcription level analyses of certain genes of interest for this study was extracted from leaf tissue. The levels of transcription obtained by RT qt PCR for the treatments were normalized by the levels of the respective controls. The threshold was set at 1.5 on the y axis. Here, the quantity of transcript for the different genes was highly cultivar-dependent (Figure 10). For the Cultivar Aragon, the transcript of Solyc06g036100, a putative LSI2-like silicon efflux transporter, was upregulated for both treatments, but only the level of transcript was significant for the Low Si treatment. This also happened for COX1, cytochrome c oxidase subunit 1. For the gene GAPDH, none of the treatments had significantly abundant transcript. In this case, however, it was upregulated for the Low Si treatment and downregulated for the High Si treatment. Regarding the gene SlHK3, *Solanum lycopersicum* hexokinase, it was significantly upregulated for both treatments, but more for the High Si treatment. The gene CAO, chlorophyllide and oxygenase, was significantly downregulated for both the treatments, more so for the High Si treatment. Considering the gene CNGC2, cyclic nucleotide gated ion channel 2, it was significantly downregulated for both treatments, in particular for the High Si treatment. Regarding the cultivar Gladis, the transcript of the gene Solyc06g036100 was not significantly abundant, however it was upregulated for the Low Si treatment and downregulated for the High Si treatment. For gene COX1, the gene was significantly downregulated by both treatments, slightly more by the high Si treatment. The gene GAPDH was significantly downregulated by the Low Si treatment, it was downregulated also by the High Si treatment but not significantly. The transcript levels for the gene SlHK3 were downregulated for both the treatments, but not significantly. Regarding the gene CAO, it was highly downregulated by the Low Si treatment. For the High Si treatment the gene was downregulated as well, but not significantly. The gene CNGC2 was significantly upregulated for the High Si treatments, and downregulated for the Low Si treatment, but not significantly in this case.

## 3. Discussion

### 3.1. Cultivar Specificity

In this work, we found a general cultivar dependence for all the parameters that we measured as being related to the Si treatment, which is typical of tomato plants [46,48]. The two cultivars chosen have already been studied for their behavior towards silicon and arsenic treatments and have demonstrated different responses to both elements [49]. This cultivar dependence is visible in all the images of the Discriminant Canonical Functions analysis in Appendix A as far as the physical traits are concerned, because in this image it is clear that the in the two cultivars the different treatments cluster among themselves and distantly form one another. In fact, in all the discriminant canonical analyses, the groups of dependent variables are positioned in different areas of the xy plane, meaning that the two canonical functions (x and y) that explain the variability for the same independent variables, but for different cultivars, are different from figure to figure (Figure 1, Figure 2, and Appendix A). In Appendix A, where the data are grouped per plant parts the set of data are more compact, indicating less difference among the plant parts within the different treatments. In Appendix A, where the data are grouped according to the treatment the data are more scattered, indicating more difference in the plant parts when grouped according to the treatment.

### 3.2. Plant Yield

Regarding the biomass, there was no increase in the biomass, fresh or dry, in respect to the control for any of the two treatments. Additionally, the number and weight of fruits were not significantly different, both in accordance, but also contrary to what is written in literature (Appendix A). In fact, in literature there is no accordance to the positive effects on the yield of tomato plants supported by Si fertilization, and according to some literature the administration of Si increases the biomass. According to other literature it leaves it as it is, but it is all still cultivar-dependent [50,51,52].

### 3.3. Silicon Concentration and Distribution in Plant Cells

The concentrations of Si varied markedly in the two cultivars, while in Gladis it was always lower than the control, as the treatment caused an increase in Si in the roots of Aragon, confirming that this cultivar absorb more Si in respect to the other, which was also visible from the µ-XRF distribution (Figure 5 and Appendix A) [49]. Aragon thus appeared more responsive to Silicon treatments than Gladis. Regarding the distribution of the element within the cells, it is also different because in the fruit of Aragon the concentration of Si within the cells appears higher than in Gladis (Figure 6 and Figure 8). However, in both cases they appear slightly higher than in controls. For the leaf of Aragon treated with the high concentration of Si, the distribution of Si was comparable between the two cultivars, as was the concentration (Figure 5c,d and Appendix A). The fact that both tomato cultivars had more Si in the roots agrees with their classification of tomato plants as low Si uptakers made by Ma and Yamajy (2006) [53] and Hodson et al. (2005) [46]. The visualization of the distribution of the elements within the cells was pivotal to understanding the different responses of the two cultivars to the different treatments. In fact, discovering that Si does penetrate within the cells and remains attached to the cell wall in the High Si treatment helped us to discriminate the differences between the two cultivars (Figure 6, Figure 7, Figure 8 and Figure 9 and Appendix A).

### 3.4. Antioxidant Activity of Si in Plants

It is well-known that Si exerts a protectant activity against biotic and abiotic stresses [54,55,56]. Regarding the abiotic stress, it has been established that the beneficial activity of Si is caused by its ability to induce the antioxidative defense of the plant, both enzymatic and non-enzymatic [57,58,59]. In our case, we found that the two cultivars had different induction of enzymes according to the treatment, but the higher treatment never induced the stronger reaction (Figure 3 and Figure 4 and relative Tukey’s tables). In fact, H_2_O_2_ did not diminish with the increase of Si treatment in any circumstances (Figure 3b and Figure 4b). On the contrary, it increases. In fact, COX1, GAPDH, and CAO, the three genes mostly devoted to increasing ROS activity, were downregulated by the treatments (Figure 10). Clearly, the treatments with Si must have activated other genes for the production of ROS, in particular H_2_O_2_, that we have not taken into consideration This trend is also followed by the non-enzymatic defense such as the production of proline, which is accumulated as a common physiological response to various stresses, osmotic and oxidative [45,60], or of Phenols [61], which serve as protectors against ROS as they are ROS chaperones [47,62] (Figure 3a and Figure 4a,d and relative Tukey’s tables). Additionally, the malondialdehyde (MDA), which gives a measure of the lipid oxidation by the ROS, did not follow a statistically significant trend different from treatment to treatment, but only different within the plant parts. The trend followed by all the antioxidant enzymes was strictly cultivar-dependent. The most responsive cultivar was Aragon, while Gladis was less influenced by the treatments. The difference in the response to the Silicon treatments in two cultivars could be ascribed to their genetic background that allows them to absorb different quantities of Si, independently from the intensity of the treatment. The fact that the High treatment had, in general, a lower response compared to the Low Si treatment and to the control could be due to the formation of a plaque of Si on the roots formed by the precipitation of silicic acid on the cell walls, forming cell wall suberization [20,37,47,63].

### 3.5. Chemical Composition of Si-Treated Plants

In the rhizosphere, Si amelioration of metal toxicity in plants has been explained by the Si-mediated metal precipitation in the growth media, which decreases the toxic metal availability for plants in soils [20,37,47,63]. However, in some crops, metals are immobilized to nontoxic forms inside roots or leaves as a plant mechanism to decrease metal toxicity [64,65]. Furthermore, a different metal distribution inside the plant due to the Si presence is overall evidence for the micronutrients under toxicity conditions [66]. Silicon may ameliorate plant micronutrient deficiency symptoms through a change of metal distribution inside the plant (Table 1) [51,67,68]. However, we did not observe drastic changes in the concentrations of macro and micronutrients in our experiment (Appendix A and all relative Tukey’s tables). Interestingly, the interaction between treatments and plant parts for both the cultivars holds only for few elements: Mn, P, Mg, and not for Si or other macronutrients as Ca or micronutrients as Fe (Table 1). Therefore, in our case, Silicon treatments did not have any interesting effects on the elemental pool in the plant’s parts in both cultivars, especially in Gladis, which was demonstrated to be less responsive to the Si treatments.

### 3.6. Molecular Mechanisms in Si Treatment

In some studies, the molecular response to Si application has been exanimated when plants were in conditions of stress, but never when the plants were not stressed [69,70,71,72]. The study of the yield of tomato plants under the different Si treatments can be found in Appendix A. The Solyc06g036100, putative LSI2-like silicon efflux transporter, was upregulated significatively only in Aragon under the treatment with Low Si, which means that that condition was the only one in which there was putative active transport in the plant leaf [73]. Additionally, COX1 (Cytochrome c oxidase I), which regulates energy and carbohydrate metabolism in the mitochondria, was upregulated in the same cultivar under the same treatment, which means that those are good conditions for the functioning of mitochondria, while in Gladis it was downregulated significatively for both treatments, indicating that High and Low Si treatment hindered mitochondria activity (Figure 10) [74]. GAPDH ((NAD-dependent glyceraldehyde-3-phosphate dehydrogenase), when upregulated, induces excessive ROS, and was downregulated in all treatment except in Aragon with the Low Si treatment, even though significatively only for Gladis with the Low Si treatment (Figure 10). This protein downregulation was, in general, related to downregulating the expression of Calvin cycle-related genes and the synthesis of chloroplast proteins [75,76]. Sl HXK3, Hexose phosphorylation, is an essential step of sugar metabolism. Only two classes of glucose and fructose phosphorylating enzymes have been found in plants. Tomato is the only plant from which four HXK and four FRK genes have been identified and characterized [77]. Here, only the Aragon RNA corresponding to the treatment with the highest concentration of Si was overabundant (Figure 10). It has been recently discovered that the inhibition of mtHXK activity can contribute to the mitochondrial ROS production [78]. CAO (Chlorophyllide and oxygenase) catalyzes the formation of Chlorophyllide b. In our work it was downregulated in all the treatments except Gladis, which was treated with the high concentration of Si (Figure 10). This could mean that there is a downregulation in the chloroplast of the production of the chlorophyll, which is contrary to what is usually maintained about the function of Si in stressed plants [36,79]. It is likely in this case that the excess of Si had a negative effect on photosynthesis, while in the case of High Si treatments, where Si remained confined in the roots, the aerial parts did not suffer from Si excess. CNGC2 (Nucleotide-Gated Channel 2) encodes a protein with Ca^2+^ influx channel activity and is expressed in the leaf surrounding the free endings of minor veins, which is the primary site for Ca^2+^ unloading from the vasculature and influx into leaf cells. In our study, only the high silicon treatment upregulated this gene with the possible consequential increase of Ca in roots, which we did not observe, but also due to the scant overabundance of the transcript (Figure 10). New studies indicate that CNGC2 is likely to have no direct role in leaf development or the hypersensitive response but, instead, that CNGC2 could mediate Ca^2+^ influx into leaf cells [80].

In this work, we joined together different types of analyses to study the effect of Low and High Si treatment on tomato plants under no type of stress. It is the first time that this type of study was carried out, and the results reveal that not only is the effect of Si on tomato plants highly cultivar-dependent, but it does not depend on the intensity of the treatment. High or low concentrations of silicon on pristine plants has, according to the cultivar, on the genomic resources available, different effects at physiological, transcriptomic, and chemical levels with only few commonalities. In particular, Si has a tendency to stay attached to the cell walls. The utilization of X-ray Fluorescence synchrotron radiation-based techniques has allowed for visualization of the distribution of Si within the cells of fruits and leaf tissues. Enzymatic and molecular analyses have revealed that oxidative stress is not of major concern when the plants are not under biotic or abiotic stress. Treatment with Si might have a sort of priming effect on the plant according to their responsiveness to Si, in this case cv. Aragon was more sensitive than cv. Gladis. We must admit, however, that the topic is still not completely elucidated in several points, and warrants further investigation.

## 4. Materials and Methods

### 4.1. Reagents and Standards

All reagents and standards were purchased from Sigma-Aldrich (St. Louis, MO, USA) unless stated otherwise.

### 4.2. Soil and Growing Conditions

Soil composition was 20% sand and 80% black peat and wood fiber (Ecomix, Vialca S.R.L., Uzzano, Italy). The soil was homogenized, passed through a 5 mm sieve, sterilized by baking at 120 °C for 1 h, then held at 50 °C for around 72 h until a constant weight had been attained. Seedlings were raised directly from uncoated seeds for 4 weeks in small pots under a 14 h photoperiod provided by 300 μmol m^−2^ s^−1^ metal halide lamps, with a 23/16 °C day/night temperature and a constant relative humidity of 50%. They were then transplanted into 4.5 L pots and irrigated with 2 L tap water (pH 7.5, EC 0.6–0.7 dS m^−1^) every week. The soils’ EC and pH were monitored following the EPA method 9045D and 9050A.

Pots were fertilized weekly by adding 200 mL of 2% *w*/*v* blood meal (Guaber S.R.L., Bologna, Italy). Plants were raised in a greenhouse providing a day/night temperature of 25–30/13–16 °C, with the natural light supplemented by 14 h per day of 300 μmol m^−2^ s^−1^ light provided by metal halide lamps. Soil, tap water, and blood meal were sampled at the beginning and at the end of treatments.

### 4.3. Si Treatments

The two cultivars were Aragon and Gladis. Each cultivar and treatment type were represented by four plants, plus four untreated plants. The treatments were initiated after the emergence of the fourth leaf and the transplantation in 4.5 L pots. Seeds and soil were the same as in Marmiroli et al. (2014) [49].

The three treatments were non-treated (NT), 3.2 mM Si (Si Low), and 16 mM Si (Si High). The nt plants were watered for 8 weeks with 2 L of water with no supplementation of Si. For the Si treatment, each pot was watered for 8 weeks with 2 L of 0.1 mM CaSiO_3_ (0.8992 g of Si) or 0.5 mM CaSiO_3_ (1.609 g of Si) for the low or high treatment, respectively. CaSiO_3_ is 10 mg 100 mL water-soluble at 20 °C. At the concentration of 2 mg L^−1^ at room temperature, the salt was completely solubilized. CaSiO_3_, also known al Wallastonite, is among the more frequently used source of Si in the field [45].

### 4.4. Sampling and Morphological Analysis

Samples were taken after 8 weeks of treatment. The whole root system was collected (principal and lateral roots), washed in deionized water, and length and weight were determined. Fruits were counted, weighed, and subdivided according to the ripening stage (green, breaker and red). Fruits at the same developmental stage, according to ‘days after flower anthesis and color’ and positioned between the 6th and 8th leaf nodes, were selected for analysis, three to five per plant [8]. Fruits were washed in deionized water: the pericarp and cuticle were retained, while the placenta and seeds were discarded. The aerial part was measured for length and weight and middle leaflets of the same age and positioned between the 6th and 8th nodes up the stalk were collected for analysis, three to four according to the cultivar. Samples of the four specimens were grouped together before averaging.

According to the analysis performed, fruits, leaves and roots were oven-dried or snap frozen in liquid nitrogen and stored at −80 °C until use, or assayed immediately after harvest.

### 4.5. Silicon (Si) Content Determination

Inductively coupled plasma optical emission spectrometry (ICP-OES) was employed to determine Si content of fruits, and Si extraction followed van der Vorm (1987) [81], with modifications. Plant material was dried and powdered; a 300 mg sample was reduced to ash in a muffle furnace for 3 h at 550 °C. The ash was suspended in 12.5 mL 0.08 M H_2_SO_4_ (Carlo Erba, Milan, Italy), and added with 0.5 mL 23 M HF (Acros Organics, Geel, Belgium); the suspension was shaken for 1 h, then left overnight. The Si content of the resulting solution was measured by ICP-OES using an Optima 7300 DV device (Perkin Elmer, Waltham, MA, USA). All analyses were performed in triplicate. The instrument parameters were set as follows: power 1.4 kW; plasma gas flow rate 15 L min^−1^; nebulizer gas flow rate 0.78 L min^−1^; auxiliary gas flow rate 0.2 L min−1148; sample flow rate 0.85 mL min−1149; Si wavelengths 251.619 nm and 212.422 nm. A calibration curve was prepared from a Si standard solution (Perkin Elmer, Waltham, MA, USA), and was used to convert the sample absorbances into Si concentrations.

### 4.6. Plants Chemical Analyses

Plant (~0.25 g) trace elements were extracted with 1 mL 30% H_2_O_2_ + 9 mL concentrated HNO_3_ using microwave-assisted pressure digestion (Mars Xpress, CEM, NC, USA). Trace element concentrations in plants were analyzed by inductively coupled plasma–mass spectrometry (XSERIES 2 ICP-MS; Thermo Scientific, MA, USA). The accuracy was routinely checked by reference to international certified standard water (NWRI-TMDA−62) and to digests of standard reference plant material (NCS DC73349) using microwave-assisted pressure digestion, and their concentrations were determined by ICP-MS as described above. Accuracy was checked by referencing digests of standard reference [82].

### 4.7. Spectrophotometric Analysis

All the spectrophotometric analyses were performed using a Beckman D640 spectrophotometer (Roma, Italia).

### 4.8. H_2_O_2_ Content

H_2_O_2_ was quantified via a colorimetric method, following Junglee et al. (2014) [83]. The leaves and fruits were snap-frozen in liquid nitrogen and ground to a powder. A 100 mg sample was extracted in 1 mL of one part 0.1% (*w/v*) trichloroacetic acid (TCA) (Honeywell Riedel-de Haen^®^, Seelze, Germany), one part 10 mM PBS (phosphate buffered saline) (pH 7), and two parts 1 M KI. The homogenate was centrifuged (12,000× *g*, 4 °C, 15 min) and the supernatant was held for 20 min at room temperature, after which the absorbance was read at 390 nm. The absorbances were converted into H_2_O_2_ concentrations via a standard curve based on a chemical grade H_2_O_2_ preparation.

### 4.9. Lipid Peroxidation

The extent of lipid peroxidation was assayed using the thiobarbituric acid test, which assays for the presence of malondialdehyde (MDA). Following Murshed et al. (2014) [84], leaves and fruits were snap-frozen in liquid nitrogen, ground to a powder, and a 200 mg sample of plant material was suspended in 1 mL 0.1% (*w*/*v*) TCA (Honeywell Riedel-de Haen^®^, Seelze, Germany) and centrifuged (12,000× *g*, 15 min). A 0.5 mL aliquot of the supernatant was added to 1 mL 0.5% (*w*/*v*) thiobarbituric acid (TBA) in 20% (*w*/*v*) TCA (Honeywell Riedel-de Haen^®^, Seelze, Germany) and held at 95 °C for 30 min, after which the reaction was quenched by immersion in an ice bath. After a brief vortex, the absorbance of the supernatant was read at 532 nm. The absorbances were corrected by subtracting the reading made at 600 nm. The absorbances were converted into MDA contents via a standard curve based on a commercial preparation of MDA.

### 4.10. Total Phenolic Content

The total phenolic content (TP) was measured on methanol extracts of leaves and fruits prepared according to Capanoglu et al. (2008) [85]. Tissues were snap-frozen in liquid nitrogen and ground to a powder. A 100 mg aliquot was combined with 1 mL 75% methanol and sonicated (Transsonic T460, Elma Schmidbauer GmbH, Singen, Germany) for 15 min at 35 kHz, then centrifuged (1000× *g*, 4 °C, 10 min). The pellet was subjected to a second round of extraction, and the two resulting supernatants were pooled. The extracts were stored at −20 °C until use.

Following Singleton and Rossi (1965) [86], a 100 µL aliquot of the methanolic extract was mixed with 0.75 mL Folin-Ciocalteu reagent (a mixture of phosphomolybdate and phosphotungstate) and allowed to stand at 22 °C for 5 min, after which 0.75 mL 60 g L^−1^ sodium bicarbonate was added. After 90 min at 22 °C, the absorbance was read at 725 nm. The total phenolic content was calculated from a calibration curve using gallic acid (GA) as the standard. Total phenolic contents are given as µg GA equivalent per g tissue (fresh weight).

### 4.11. Proline Content

The proline content was determined following Chen and Zhang (2016) [87]. Plant tissue (0.5 g) was ground to a powder in liquid nitrogen and homogenized in 5 mL of 3% sulfosalicylic acid (VWR International, Radnor, PA, United States). The homogenate was centrifuged at 14,000× *g* for 10 min at 4 °C and the supernatant was collected. An aliquot of 50 μL of the protein extract was mixed to a reaction buffer containing 3% sulfosalicylic acid, glacial acetic acid, and 2.5% ninhydrin (VWR International, Radnor, PA, USA) solution (1:1:2). Ninhydrin was dissolved in 100 mM phosphate buffer saline (pH 7) and glacial acetic acid 1:1.5. After 15 min of incubation at 100 °C, the reaction was stopped in ice and the absorbance was read at 520 nm. The proline content was calculated from a calibration curve of L-proline and expressed as proline µg per g tissue (fresh weight).

### 4.12. Antioxidant Enzymes Assays

Frozen fruits (1 g) were grounded by liquid nitrogen and homogenized in 5 mL of 50 mM sodium phosphate buffer (pH 7.0), containing 0.1 mM ethylenediaminetetraacetic acid (EDTA), 1 mM ascorbic acid, and 1% PVPP [88]. The homogenate was centrifuged at 12,000× *g* for 15 min at 4 °C.

Frozen leaves and roots (0.5 g) were grounded by liquid nitrogen and homogenized in 2 mL of 50 mM Tris-HCl buffer (pH 7.8), containing 0.1 mM EDTA, 0.2% Triton X-100, 1 mM PMSF, and 2 mM DTT, following Çelik et al. (2017) [89]. The homogenate was centrifuged at 12,000× *g* for 30 min at 4 °C.

The supernatants were frozen at −80 °C and then used to evaluate the activity of superoxide dismutase (SOD), catalase (CAT), peroxidase (POD), and ascorbate peroxidase (APX).

The SOD activity was determined by measuring its ability to inhibit the photochemical reduction of nitro blue tetrazolium chloride (NBT) according to Beauchamp and Fridovich (1971) [90]. The enzymatic extract (100 μL) was added to a reaction buffer containing 50 mM sodium phosphate buffer (pH 7.8), 33 μM NBT, 10 mM L-methionine, 0.66 mM EDTA, and 3.3 μM riboflavin. After 10 min of incubation, the absorbance was read at 560 nm. The percentage of NBT reduction inhibition was calculated as [(OD_control_ − OD_treated sample_)/OD_control]_∗100. One unit of SOD activity was defined as the amount of enzyme that caused 50% inhibition of NBT reduction.

The CAT activity was evaluated as disappearance of H_2_O_2_ at 240 nm, followed for 5 min according to Çelik et al. (2017) [89]. An aliquot of 100 µL was added to a reaction buffer containing 50 mM sodium phosphate buffer (pH 7) and 30 mM H_2_O_2_. CAT was calculated by using the molar extinction coefficient of 39.4 mM^−1^ cm^−1^ [91].

The POD activity was measured by the increase in absorbance at 470 nm followed for 5 min, according to Çelik et al. (2017) [89]. The enzymatic extract (100 μL) was added to a reaction buffer containing 20 mM sodium acetate (pH 5) (J.T. Baker, Deventer, Holland, The Netherlands), 20 mM guaiacole, and 30 mM H_2_O_2_. POD was calculated by using the molar extinction coefficient of 26.6 mM^−1^ cm^−1^.

The APX activity was evaluated as the decrease of absorbance at 290 nm due to the H_2_O_2_-dependent oxidation of ascorbate [92]. The reaction mixture consisted of 50 mM sodium phosphate buffer (pH 7), 0.1 mM EDTA, 0.25 mM ascorbate, 1 mM H_2_O_2_, and 100 μL of enzymatic extract. After 5 min at 25 °C the absorbance was followed for 5 min. APX was calculated by using the molar extinction coefficient of 2.8 mM^−1^ cm^−1^ [93].

Frozen fruits, leaves, and roots (1 g) were grinded in liquid nitrogen and homogenized in 2 mL of ice-cold buffer containing 0.1 M sodium phosphate buffer (pH 6.5), 20% glycerol, 14 mM dithiothreitol (DTT), 1 mM phenylmethylsulfonylfluoride (PMSF), and 1 mM EDTA [94]. Homogenates were centrifuged 20 min at 15,000× *g* (4 °C). Supernatants were collected, stored at −80 °C, and then used to evaluate the activity of glutathione s-transferase (GST) and glutathione peroxidase (GPX).

The GST activity was measured following the Habig and Jakoby (1981) [95] methodology. A reaction buffer containing 50 mM sodium phosphate buffer (pH 7.5), 50 mM glutathione (GSH), and 100 mM 1-chloro-2,4-dinitrobenzene (CDNB) was prepared and incubated 5 min before adding 100 μL of enzyme extract. The absorbance at 340 nm generated by the conjugation of 1 mM GSH with 1 mM of CDNB was monitored during 5 min at 25 °C [94]. The GST activity was calculated by using the molar extinction coefficient of 9.6 mM^−1^ cm^−1^.

The GPX activity was determined following Rotruck et al. (1973) [96]. A reaction mixture containing 50 mM sodium phosphate buffer (pH 7), 25 mM sodium azide, 5 mM GSH, 1.25 mM H_2_O_2_, and 50 μL of enzyme extract was incubated 10 min at 37 °C. The reaction was stopped adding 10% TCA (Honeywell Riedel-de Haen^®^, Seelze, Germany) and centrifuged for 10 min at 2000 rpm. An aliquot of supernatant was added (1:1: 0.5 *v*/*v*) to 0.4 M Na_2_HPO_4_ and 1 mM 5,5′-Dithiobis-(2-Nitrobenzoic Acid) (DTNB) and incubated 10 min at 37 °C. The decrease in absorbance at 412 nm of the colored complex GSH-DTNB was monitored during 5 min. The GPX activity was calculated by using the molar extinction coefficient of 13.6 mM^−1^ cm^−1^.

Extraction of tomato lipoxygenases was carried out according to Hu et al. (2011) [97]. Fruits, leaves, and root were grinded in liquid nitrogen and 0.2 g of powder were homogenized with 1 mL of 50 mM sodium phosphate buffer (pH 7) and 1.5% (*w*/*v*) Triton X−100. Samples were centrifuged 15 min at 14,000× *g* and the supernatants were collected, stored at −80 °C, and assayed for LOX activity.

The LOX activity was evaluated in a reaction mixture containing 50 mM sodium phosphate buffer (pH 7), 2.5 mM linoleic acid, and 0.1% Tween 20 (*w*/*v*). The absorbance at 234 nm derived from the conjugated diene chromophore of fatty acid hydroperoxides was monitored for 5 min. The LOX activity was calculated by using the molar extinction coefficient of 25 mM^−1^ cm^−1^.

### 4.13. RNA Extraction and Transcriptome Analysis

Total RNA was extracted from 0.1 g of frozen plant material using a Sigma-Aldrich Spectrum Plant Total RNA Kit (Sigma-Aldrich, St. Louis, MO, USA). Three biological replicates per treatment were extracted. The total RNA quality and quantity were assessed by gel electrophoresis and by determining the 260/280 ratio with a Thermo Scientific Nanodrop Lite Spectrophotometer (Thermo Fisher Scientific, Wilmington, DE, USA). A 1 μg aliquot of total RNA was reverse-transcribed in cDNA using a QuantiTect^®^ Reverse Transcription kit (Qiagen). The synthesized cDNA was used for gene amplification (Appendix A). In total, six genes related to stress response, Si and Ca transport, drought tolerance, and antioxidant enzymes were identified in each sample. The expression level of Actin was used as internal control. The qRT-PCR was based on Power SYBR^®^ Green PCR Master Mix (Life Technologies, Carlsbad, CA, USA), following the manufacturer’s instructions, and was implemented in an Applied Biosystems 7900HT Fast Real-Time PCR System device (Applied Biosystems, Foster City, CA, USA). Three independent experiments were carried out, each performed in triplicate. The specificity of each of the amplicons was determined from a melting curve. The relative abundance of each transcript was obtained using the ΔΔCt method [98], and fold transcription changes (RQ) were calculated from the expression 2^−ΔΔCt^.

### 4.14. Low Energy µ-XRF (LE µ-XRF)

Tomato plants were sampled as already described. Parts of roots, fruits, and leaves were cut and submerged in glutaraldehyde tri-phosphate for fixation. Afterwards, samples were dehydrated in gradients of alcohol (30%, 50%, 80%, 96%, 100%) and embedded in epoxy resin Technovit 7100 (Bio Optica Milano s.p.a, Italy) following the manufacturer’s instructions. The specimen blocks were sectioned with an automated rotary microtome (Slee Medical GmbH, Mainz, Germany).

The samples were prepared and analyzed according to Marmiroli et al., 2020 [99]. Briefly, after being mounted on a sample holder, the samples were inserted in the vacuum chamber of the beamline TwinMic [100] at Trieste Synchrotron Elettra. For the µ-XRF analyses, the TwinMic microscope was utilized in scanning transmission mode (SXM), the beam was focused on the sample through a zone plate (diameter: 600 μm; outermost zone width: 50 nm), and a micrometric probe of 1.2 µm diameter was delivered [101]. Samples were raster-scanned perpendicularly to the incoming monochromatic beam, while a fast readout CCD camera collected the transmitted X-rays, and an 8-silicon drift detector-based XRF system acquired the emitted fluorescence photons [102]. The obtained absorption and phase contrast images outline the morphological sample features at sub-micrometer length scales, while the simultaneous detection of the low energy µ-XRF correlates the elemental distribution to the morphology. Elemental distribution has been obtained with PyMCA software [103] by deconvolving and fitting the XRF spectra. A photon energy of 2 keV was used to excite and obtain optimal emission conditions for the elements of major interest (Si, Al, Mg, Na, and O) with a spot size of 1.2 μm, a dwell time of 8 s per pixel for XRF mapping, and a CCD dwell time of 50 ms per SXM imaging. Each map lasted approximately 5–7 h, depending on the dimensions of the scanned area.

The elemental maps shown in Figure 5, Figure 6, Figure 7, Figure 8 and Figure 9 were generated in the following way: each group of maps for every treatment/type of organ is not homogeneous in terms of intensity counts within itself, thus it needs to be rescaled with all the other maps in terms of photons per seconds acquired by the detector. Then, in the graphical representation, the color code was rescaled as homogeneously as possible to give the possibility to compare the different groups of maps as much as possible.

### 4.15. Statistics

The analysis of variance (ANOVA), both one way and two ways, followed by the Tukey’s post hoc test, whose *p* values are reported in the Appendix A the Discriminant Functions Analysis and the two-way ANOVA, is reported in Table 1 where *p* < 0.001 are in Bold, *p* < 0.005 are in Italics, and *p* < 0.01 are in Italics Underscored, were all performed with IBM SPSS v.27.

## Figures and Tables

**Figure 1 ijms-23-15837-f001:**
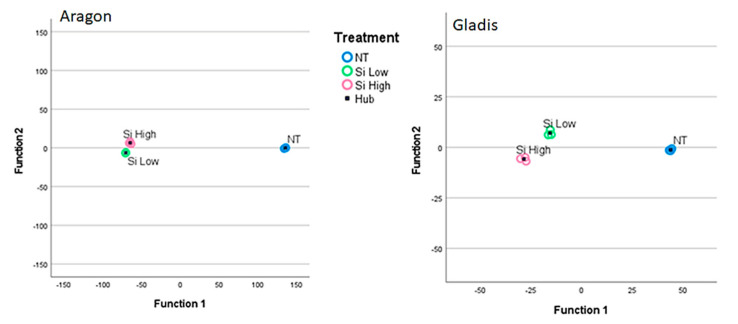
Discriminant canonical functions analysis for the physiological analyses according to treatments (Control, Si Low, Si High) and grouped by parts of the plant: blue = leaf, green = root, pink = fruit. Blue square = hub of the group. Left side Aragon, right side Gladis.

**Figure 2 ijms-23-15837-f002:**
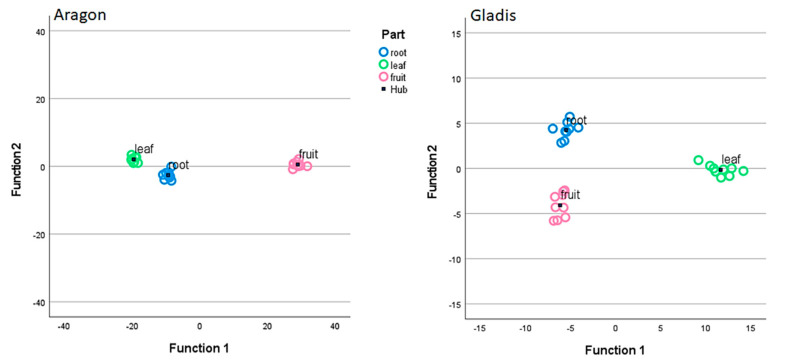
Discriminant canonical functions analysis for the chemical analyses according to treatments (Control, Si Low, Si High) and grouped by parts of the plant: blue = leaf, green = root, pink = fruit. Blue square = hub of the group. Left side Aragon, right side Gladis.

**Figure 3 ijms-23-15837-f003:**
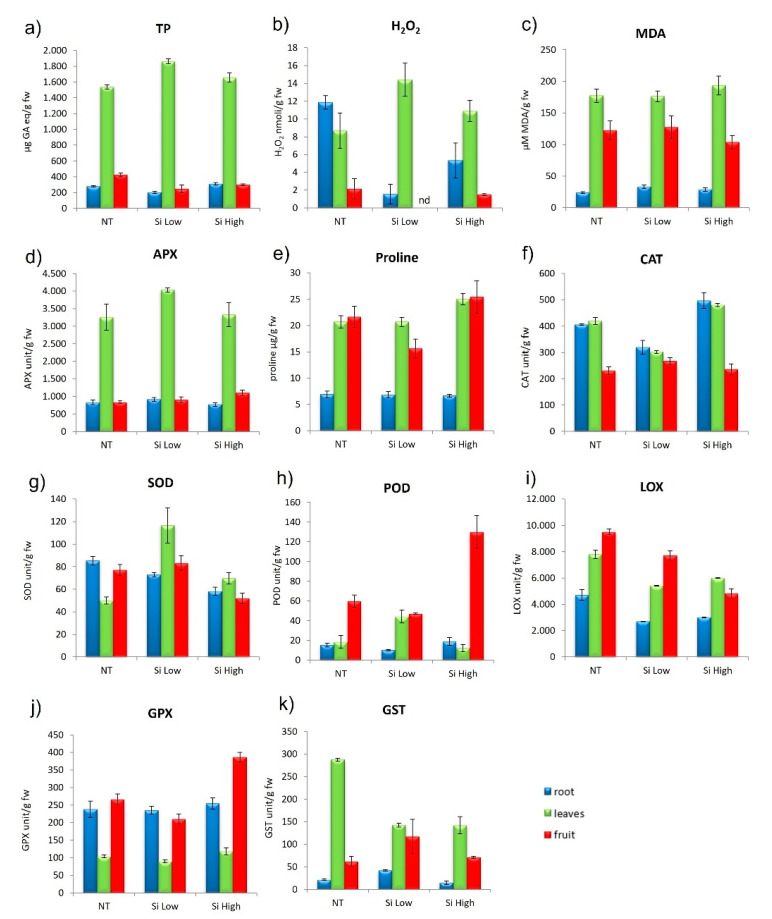
Effect of Si Low and Si High physiological parameters in cvs. Aragon: (**a**) total phenolic (TP), (**b**) hydrogen peroxide (H_2_O_2_), (**c**) lipid peroxidation (MDA), (**d**) ascorbate peroxidase (APX), (**e**) proline, (**f**) catalase activity (CAT), (**g**) superoxide dismutase activity (SOD), (**h**) peroxidase activity (POD), (**i**) lipoxygenase (LOX), (**j**) glutathione peroxidase (GPX), (**k**) Glutathione S-transferases (GST). Values equal to 0 means below detection limit (nd). ANOVA was performed followed by Tukey’s HSD test. Tukey’s *p* values are explicated in Appendix A.

**Figure 4 ijms-23-15837-f004:**
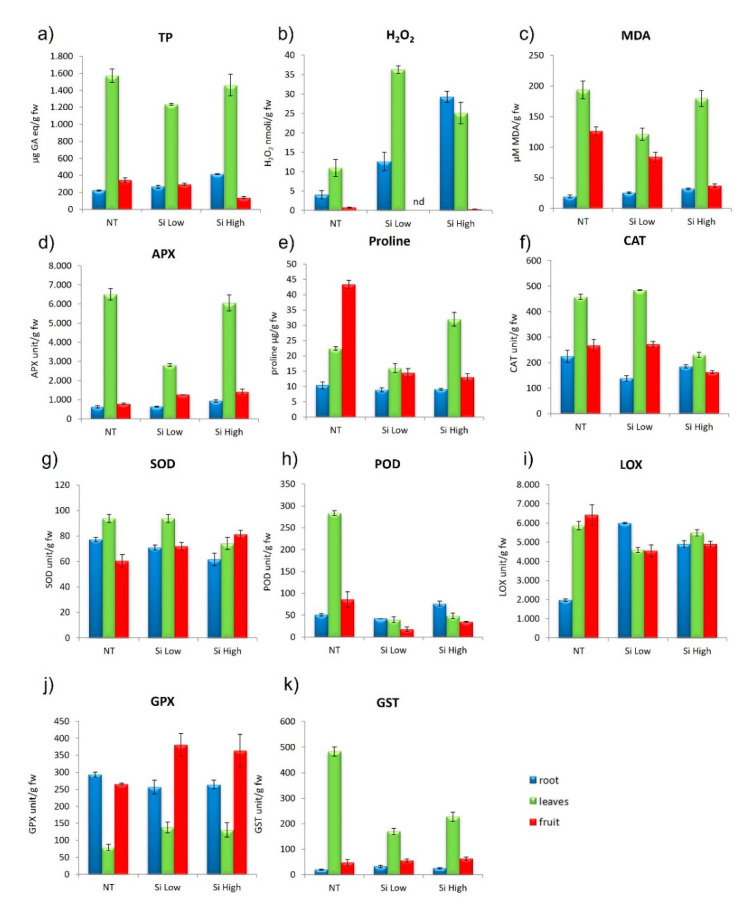
Effect of Si Low and Si High physiological parameters in cvs. Gladis: (**a**) total phenolic (TP), (**b**) hydrogen peroxide (H_2_O_2_), (**c**) lipid peroxidation (MDA), (**d**) ascorbate peroxidase (APX), (**e**) proline, (**f**) catalase activity (CAT), (**g**) superoxide dismutase activity (SOD), (**h**) peroxidase activity (POD), (**i**) lipoxygenase (LOX), (**j**) glutathione peroxidase (GPX), (**k**) Glutathione S-transferases (GST). Values equal to 0 means below detection limit (nd). ANOVA was performed followed by Tukey’s HSD test. Tukey’s *p* values are explicated in Appendix A.

**Figure 5 ijms-23-15837-f005:**
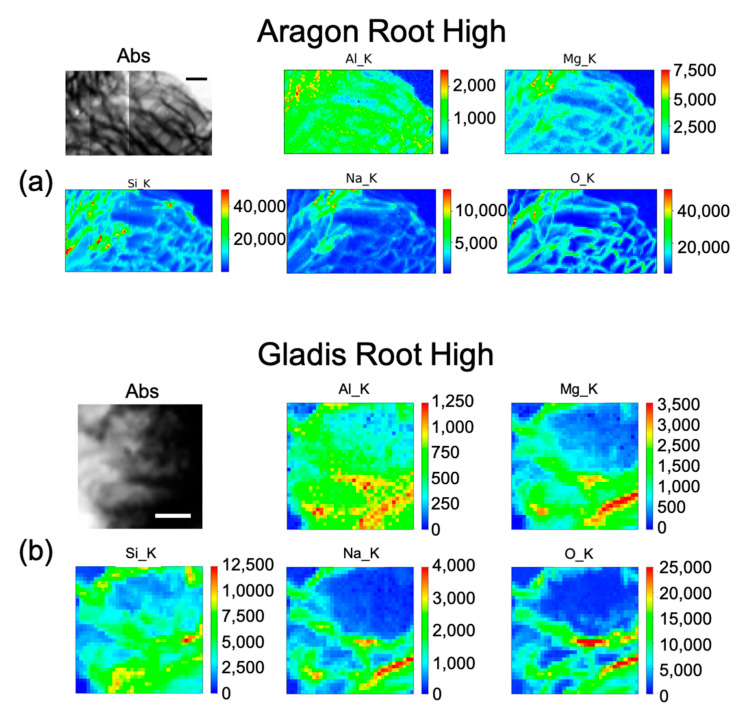
Micro-X-ray Fluorescence (µ-XRF) elemental maps of O, Al, Na, Mg, and Si, depicting the distribution within the cells in (**a**) Aragon Root High Si treatment, and of (**b**) Gladis Root High Si treatment, shown with their corresponding absorption image (Abs). Scale bar is 10 µm.

**Figure 6 ijms-23-15837-f006:**
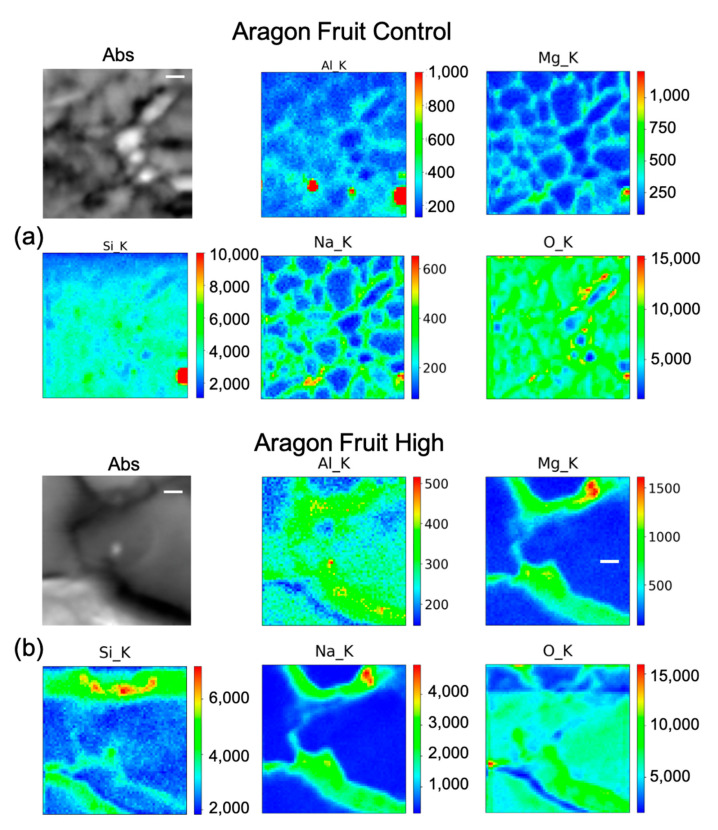
Micro-X-ray Fluorescence (µ-XRF) elemental maps of O, Al, Na, Mg, and Si, depicting the distribution within the cells of (**a**) Aragon Fruit Control, and of (**b**) Aragon Fruit High Si treatment, shown with their corresponding absorption image (Abs). Scale bar is 10 µm.

**Figure 7 ijms-23-15837-f007:**
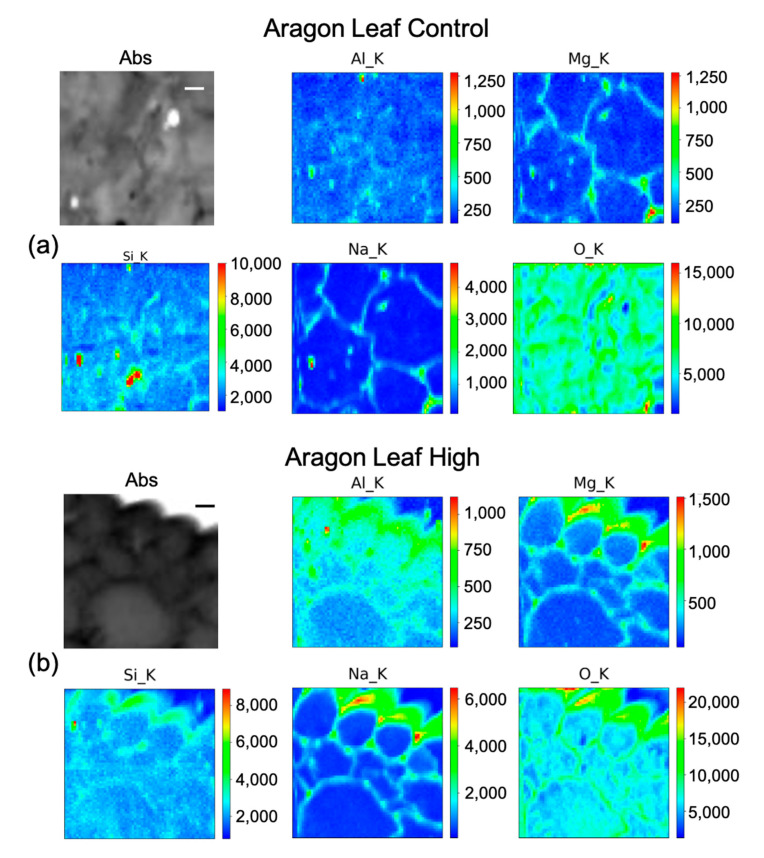
Micro-X-ray Fluorescence (µ-XRF) elemental maps of O, Al, Na, Mg, and Si, depicting the distribution within the cells of (**a**) Aragon Leaf Control, and of (**b**) Aragon Leaf High Si treatment, shown with their corresponding absorption image (Abs). Scale bar is 10 µm.

**Figure 8 ijms-23-15837-f008:**
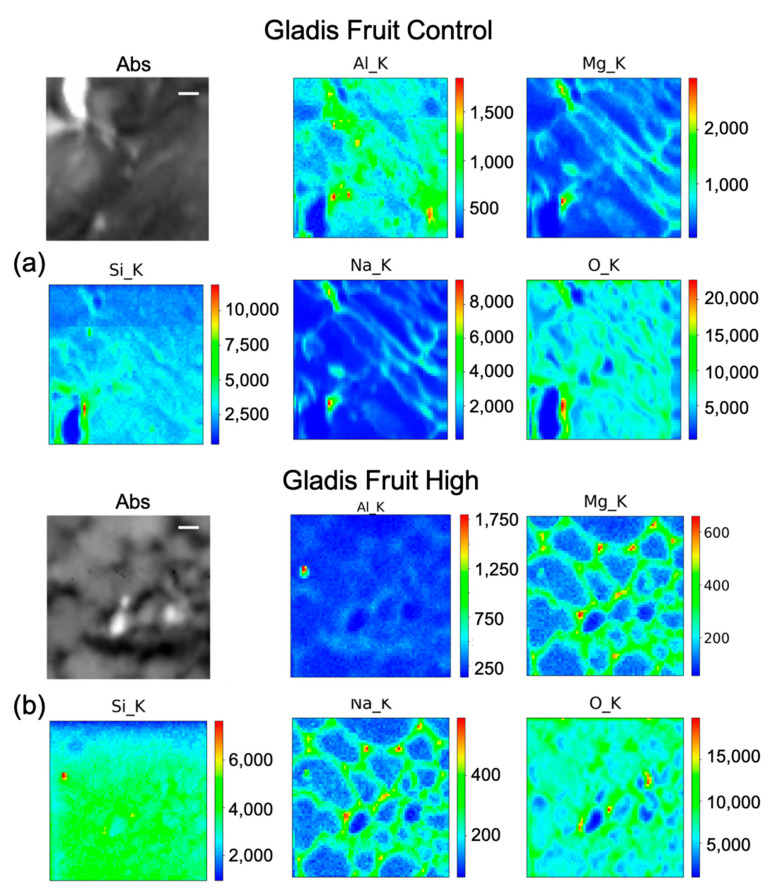
Micro-X-ray Fluorescence (µ-XRF) elemental maps of O, Al, Na, Mg, and Si, depicting the distribution within the cells of (**a**) Gladis Fruit Control, and of (**b**) Gladis Fruit High Si treatment, shown with their corresponding absorption image (Abs). Scale bar is 10 µm.

**Figure 9 ijms-23-15837-f009:**
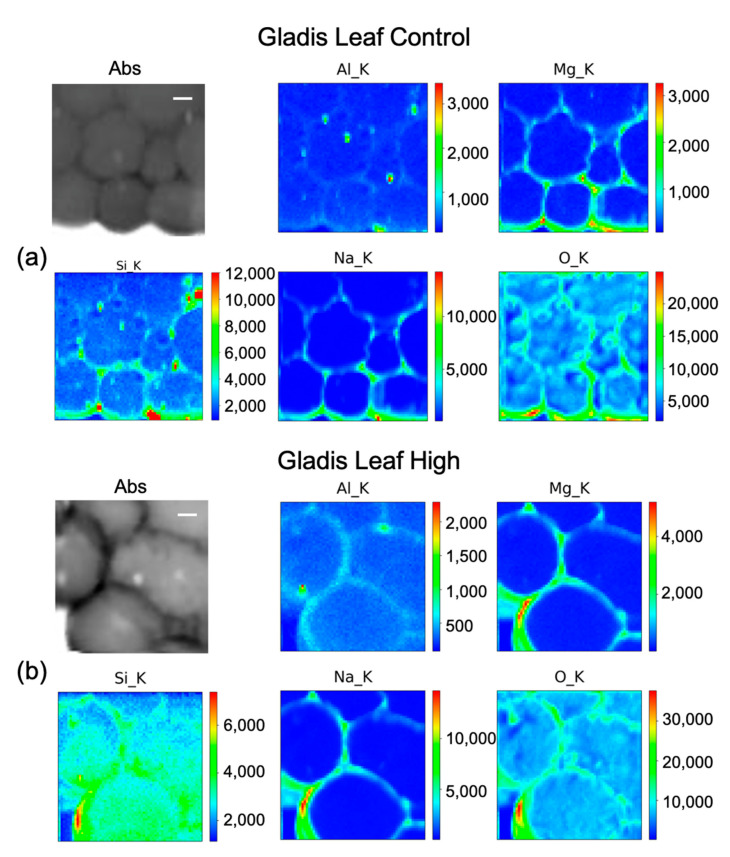
Micro-X-ray Fluorescence (µ-XRF) elemental maps of O, Al, Na, Mg, and Si, depicting the distribution within the cells of (**a**) Gladis Leaf Control, and of (**b**) Gladis Leaf High Si treatment, shown with their corresponding absorption image (Abs). Scale bar is 10 µm.

**Figure 10 ijms-23-15837-f010:**
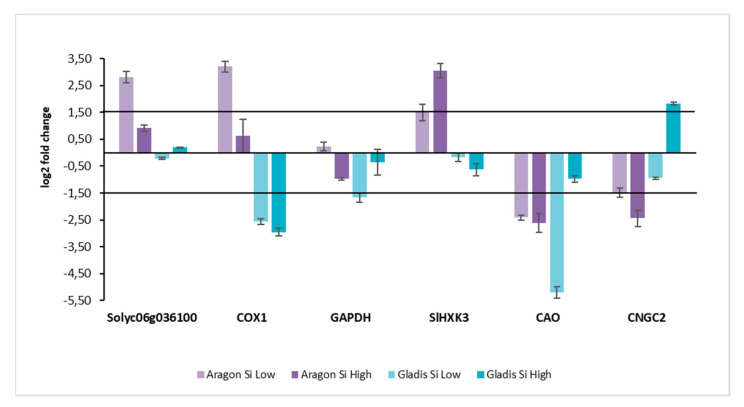
Genes expression values obtained by RT-PCR. Color bars: light purple Aragon treatment Si Low; Dark Purple Aragon treatment Si High; Light cyan Gladis treatment Si Low; Dark cyan Gladis treatment Si High. All the measures have been normalized with the respective Controls. Data were normalized on NT samples. Black horizontal lines: Log2 fold change threshold set at 1.5 and −1.5 for the significativity of the expression values. Genes are at the bottom, from left to right: Solyc06g036100 (putative LSI2-like silicon efflux transporter), COX1 (Cytochrome c oxidase I), GAPDH (NAD-dependent glyceraldehyde-3-phosphate dehydrogenase), Sl HXK3 (hexokinase), CAO (Chlorophyllide a oxygenase), and CNGC2 (Nucleotide-Gated Channel 2).

**Table 1 ijms-23-15837-t001:** Two-way ANOVA for the chemistry analyses of the different parts of the plants (root, leaf, fruit) treated with different concentrations of Silicon (Non-treated, Si-Low, Si-High). In the different columns the F statistics are reported, and the *p* values of the ANOVA according to plant treatments, plant parts, and their interaction for all the elements analyzed in the study. In bold *p* < 0.001; in italics *p* < 0.005; in italics underscored *p* < 0.01.

Element	Treatment F	Treatment *p* Value	Part F	Part *p* Value	Treatment Part F	*p* Value
**Aragon**						
**Fe**	0.842	0.447	23.115	**0.000**	2.739	0.061
**Al**	3.932	0.038	0.652	0.533	1.492	0.246
**Cu**	1.656	0.219	1.040	0.374	1.708	0.192
**Mn**	12.873	**0.000**	813.73	**0.000**	6.637	*0.002*
**Na**	22.006	**0.000**	334.67	**0.000**	8.550	**0.000**
**Ni**	0.570	0.575	3.953	0.038	0.718	0.590
**Zn**	2.845	0.084	2.440	0.115	0.499	0.737
**Mo**	0.732	0.495	5.710	0.012	1.227	0.334
**Ca**	0.925	0.414	8.784	*0.002*	1.264	0.320
**K**	0.257	0.776	12.050	**0.000**	4.035	0.017
**P**	2.084	0.153	11.117	*0.001*	4.754	* 0.009 *
**Mg**	1.099	0.355	97.970	**0.000**	13.151	**0.000**
**Si**	83.266	**0.000**	303.67	**0.000**	78.478	**0.000**
**Cu**	1.349	0.284	7.438	*0.004*	1.168	0.358
**Gladis**						
**Fe**	0.828	0.453	1.095	0.356	1.190	0.349
**Al**	8.059	*0.003*	100.08	**0.000**	7.684	*0.001*
**Cu**	50.482	**0.000**	321.19	**0.000**	44.280	**0.000**
**Mn**	37.324	**0.000**	59.953	**0.000**	16.217	**0.000**
**Na**	4.658	0.023	4.376	0.028	1.513	0.240
**Ni**	13.610	**0.000**	49.039	**0.000**	11.801	**0.000**
**Zn**	1.998	0.165	0.269	0.767	1.004	0.431
**Mo**	2.898	0.081	38.213	**0.000**	2.772	0.059
**Ca**	4.009	0.036	5.827	0.011	4.421	0.012
**K**	4.499	0.026	8.204	*0.003*	2.589	0.072
**P**	21.145	**0.000**	150.50	**0.000**	8.307	*0.001*
**Mg**	7.318	* 0.005 *	543.78	**0.000**	9.396	**0.000**
**Si**	1.349	0.284	7.438	*0.004*	1.168	0.358

## Data Availability

Not applicable.

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
