# Peer review of "Combination of Biochemical, Molecular, and Synchrotron-Radiation-Based Techniques to Study the Effects of Silicon in Tomato (Solanum Lycopersicum L.)"

_ijms, 2022, doi:10.3390/ijms232415837_

Round 1

Reviewer 1 Report (Previous Reviewer 1)

After the 2nd review of the original manuscript I had told ijms not to involve me any further. However, I reluctantly have agreed to their request:

You recently reviewed the original version of the manuscript entitled "Combination of Biochemical, Molecular, and synchrotron-radiation-based techniques to study the effects of Silicon in tomato (Solanum lycopersicum L.)" on 2022-10-13, which was previously rejected pending revisions.

The authors have now revised their manuscript and have responded to your comments.

In order to support a consistent peer review process, we kindly ask you to take the time to review this improved version. The latest manuscript with the revisions indicated and authors’ point-by-point responses are attached.

Please inform us within 3 days as to whether you believe the manuscript has been significantly improved and now warrants publication in ijms, or if further revisions are required.

As a whole, both text and data presentation have improved in this re-submission, by and large addressing the queries / suggestions made during the review of the original manuscript.

However, there are still two major points, which the authors tried to explain in their cover letter, but which didn’t really find their way into the manuscript:

(1)    Figures 3 & 4 (and now S4a,b & S5a,b as well)

My query (both in review rounds 1 and 2):

In the two figures a number of “letters”, and combinations thereof accompany, the bar graph data: a, b, c, d, e and f as well as ab, ac, ad, ae, af, bc, cb. What is the meaning of these? Are they outputs from the analysis software? Are they a personal annotation chosen by the authors? In either case, a proper explanation has to be given and each letter associated with a numerical value (if there is any), or a particular (statistical) meaning. Otherwise, it is next to impossible for a non-expert reader to interpret the figures and gauge any detailed significance related to the study.

Author response:

The multiple letters serve to indicate that a measure is between two values according to the Tukey HSD post hoc test that has been performed following the one-way Analysis of Variance (ANOVA) analysis of the data. The data on the roots leaf and fruit have been treated all together thus the reason for the presence of multiple letters. They can be considered as output from the SPSS software. The explanation is added in the Figure captions

Captions of related figures: 

Different superscript letters above the standard deviation bars indicate significant differences according to ANOVA followed by post hoc Tukey’s HSD test for multiple comparisons analysis (p≤0.01). Values equal to 0 means below detection limit (nd). Multiple letters indicate that a value is statistically between the values corresponding to the two letters.

Repeat of request for change in the manuscript:

In the first instance, I take the point of the authors as to the definition of ANOVA – this is/was a consequence of the way ijms sets the order of content.

However, a completely different point altogether is to properly explain how the data were treated; what certain data entries in the data plots mean; and which value they have. The authors have ignored this for the second time.

Evidently, all the figures represent the output from the SPSS software. While researchers in the field may be familiar with SPSS and its incorporated ANOVA treatment, not everybody knows nitty-gritty details. The authors state that the letters associated with the bar graph data “indicate significant differences”. What does that mean (I am repeating myself)? Is there a hierarchy in the letters – say from a to f –, being associated with different values for the probability? For example, is “a” associated with p≤0.01 (the value given in the figure caption? Or is it the difference between two consecutive letters in the alphabet? Because the statement “Multiple letters indicate that a value is statistically between the values corresponding to the two letters” actually suggest numerical values.

I quickly looked at a couple of tutorials of ANOVA in SPSS (I personally have rarely used the SPSS software package), but I could not find any explicit description of such annotation letters, although they occasionally appeared in example plots. From this I start to get the impression that the authors use ANOVA/SPSS like a black box – likely with good skill, but without detailed understanding of the significance of the some of the output the software generates.

Without any proper explanation, and numerical association of said letters, the whole thing looks rather futile to me. Therefore, an explanation / definition needs to be given. In order to avoid repetition in the captions or the related text, a good location for this explanation could be “Section 4.15 – Statistics”. In a short paragraph the authors need to describe how and with which parameters ANOVA/SPSS was used, and to provide a definition of numerical association to individual letters.

Without a decent, meaningful explanation I will NOT accept the manuscript. Not everybody is a SPSS expert who understands all software-internal notations !

(2)    mXRF

Author response to reviewer query on false-colour coding in the mXRF maps:

… Regarding the false colour code, the explanation is that each group of maps for every treatment/ type of organ is not homogeneous within itself but is re-scaled with all the other maps in terms of photons per seconds acquire by the detector. Then the graphical representation rescaled the colour code as more homogeneously as possible to give the possibility to compare the different groups of maps as much as possible (this is clearly written in the text in line 403)

Request for change in the manuscript:

Line 403 only states: … By rescaling the map, it seems that Si is present quite uniformly inside the fruit … The mere mention of “rescaling” is completely inadequate to me. Again, like in many other responses by the authors, they explain things to me, the reviewer, reasonably well but fail to transfer this into the manuscript.

In my view, the explanation given in the author response should be included in the methods segment “Section 4.14 - Low Energy µ-XRF”, probably towards the end.

A possible description could be related to the response text above (or something similar the authors may compose):

The concentration maps shown in Figures 5 to 9 were generated in the following way: each group of maps for every treatment / type of organ is not homogeneous within itself but needs to be re-scaled with all the other maps in terms of photons per seconds acquire by the detector. Then the graphical representation rescaled the colour code as homogeneously as possible to give the possibility to compare the different groups of maps as much as possible. (Here I merely transposed the above response text – adequate formulation needs to be implemented by the authors).

(3)    In the Supplement the following minor issues need to be addressed:

(a)    Figure S4b – there is no vertical axis scale for the Gladis Si low plot

(b)    Figure S5a – the x-axis annotation for Aragon leaf is different (horizontal, not 45°)

(c)    Figure S5b – the x-axis annotation for Gladis fruit is different (horizontal, not 45°)

(d)    All captions for Figures S4a / S4b / S5a / S5b: The difference between the annotation with ** (2-star) and *** (3-star) needs to be explained. The caption only states:  The asterisks near the elements indicate those that have an interaction between treatment and part of the plant.

Author Response

Answer to Reviewer 1

Comments and Suggestions for Authors

After the 2nd review of the original manuscript, I had told ijms not to involve me any further. However, I reluctantly have agreed to their request:

You recently reviewed the original version of the manuscript entitled "Combination of Biochemical, Molecular, and synchrotron-radiation-based techniques to study the effects of Silicon in tomato (Solanum lycopersicum L.)" on 2022-10-13, which was previously rejected pending revisions.

The authors have now revised their manuscript and have responded to your comments.

In order to support a consistent peer review process, we kindly ask you to take the time to review this improved version. The latest manuscript with the revisions indicated and authors’ point-by-point responses are attached.

Please inform us within 3 days as to whether you believe the manuscript has been significantly improved and now warrants publication in ijms, or if further revisions are required.

Question 1 

As a whole, both text and data presentation have improved in this re-submission, by and large addressing the queries / suggestions made during the review of the original manuscript.

Answer 1

 We thank the Reviewer 1 for having acknowledged the effort that the Authors made to satisfy his/her requests.

Question 2

However, there are still two major points, which the authors tried to explain in their cover letter, but which didn’t really find their way into the manuscript:

(1)    Figures 3 & 4 (and now S4a,b & S5a,b as well)

My query (both in review rounds 1 and 2):

In the two figures a number of “letters”, and combinations thereof accompany, the bar graph data: a, b, c, d, e and f as well as ab, ac, ad, ae, af, bc, cb. What is the meaning of these? Are they outputs from the analysis software? Are they a personal annotation chosen by the authors? In either case, a proper explanation has to be given and each letter associated with a numerical value (if there is any), or a particular (statistical) meaning. Otherwise, it is next to impossible for a non-expert reader to interpret the figures and gauge any detailed significance related to the study.

Author response:

The multiple letters serve to indicate that a measure is between two values according to the Tukey HSD post hoc test that has been performed following the one-way Analysis of Variance (ANOVA) analysis of the data. The data on the roots leaf and fruit have been treated all together thus the reason for the presence of multiple letters. They can be considered as output from the SPSS software. The explanation is added in the Figure captions

Captions of related figures: 

Different superscript letters above the standard deviation bars indicate significant differences according to ANOVA followed by post hoc Tukey’s HSD test for multiple comparisons analysis (p≤0.01). Values equal to 0 means below detection limit (nd). Multiple letters indicate that a value is statistically between the values corresponding to the two letters.

Repeat of request for change in the manuscript:

In the first instance, I take the point of the authors as to the definition of ANOVA – this is/was a consequence of the way ijms sets the order of content.

However, a completely different point altogether is to properly explain how the data were treated; what certain data entries in the data plots mean; and which value they have. The authors have ignored this for the second time.

Evidently, all the figures represent the output from the SPSS software. While researchers in the field may be familiar with SPSS and its incorporated ANOVA treatment, not everybody knows nitty-gritty details. The authors state that the letters associated with the bar graph data “indicate significant differences”. What does that mean (I am repeating myself)? Is there a hierarchy in the letters – say from a to f –, being associated with different values for the probability? For example, is “a” associated with p≤0.01 (the value given in the figure caption? Or is it the difference between two consecutive letters in the alphabet? Because the statement “Multiple letters indicate that a value is statistically between the values corresponding to the two letters” actually suggest numerical values.

I quickly looked at a couple of tutorials of ANOVA in SPSS (I personally have rarely used the SPSS software package), but I could not find any explicit description of such annotation letters, although they occasionally appeared in example plots. From this I start to get the impression that the authors use ANOVA/SPSS like a black box – likely with good skill, but without detailed understanding of the significance of the some of the output the software generates.

Without any proper explanation, and numerical association of said letters, the whole thing looks rather futile to me. Therefore, an explanation / definition needs to be given. In order to avoid repetition in the captions or the related text, a good location for this explanation could be “Section 4.15 – Statistics”. In a short paragraph the authors need to describe how and with which parameters ANOVA/SPSS was used, and to provide a definition of numerical association to individual letters.

Without a decent, meaningful explanation I will NOT accept the manuscript. Not everybody is a SPSS expert who understands all software-internal notations !

Answer 2

We thank the reviewer 1 for this observation.

In order to satisfy the concerns of the reviewer 1 about the statistic meaning of the letters, the authors eliminated the letters from all the figures and the asterisks from the x axis of figure S4 and S5. We decided to substitute the letters with the exact p values of the Tukey’s HSD test for all dependent independent variables, which are those in Figure 3 (for the cv Aragon) and Figure 4 for cv. Gladis): total phenolic (TP), hydrogen peroxide (H2O2),  lipid peroxidation (MDA),  ascorbate peroxidase (APX),  proline,  catalase activity (CAT),  superoxide dismutase activity (SOD), peroxidase activity (POD), lipoxygenase (LOX),  glutathione peroxidase (GPX), Glutathione S-transferases (GST). Also, for the chemical elements in Figures S4a, b and S5 a, b. the statistics letters have been eliminated and substituted with the exact p values for the Tukey’s HSD post hoc test. The tables with all the Tukey’s values are reported in the supplementary materials and are for all the conditions: type of treatments: NT= not treated; Si Low= low Si treatment; Si High= High Si treatment and Parts of plants: L= leaf; R= root; F= fruit. These tables are respectively: from Table S4 to Table SS7 for Figure 3 and Figure 4; From Table S8 to Table S12 for Figure S4a, b and Figure S5a,b. In this way the reader can verify the significativity of the analyses directly comparing the p values of the Tukey’s HSD test with the graphical representation, without having to interpret or to gouge the meaning of the letters. The asterisks that were in the x axis of Figure S4a,b and Figure5a,b, have been eliminated and substituted with table 1 where all the ANOVA F statistics and p values for each element with the signaficativity evidenced without asterisks, but with alternative modes which is that the P values are in Bold P <0.001 in Italics P <0,005, P<0.01 in Italics Underscored. With these changes we hope to have made the statistics easier and clearer to understand.  In the paragraph 4.15 we have explained these changes

Question 3

(2)    mXRF

Author response to reviewer query on false-colour coding in the mXRF maps:

… Regarding the false colour code, the explanation is that each group of maps for every treatment/ type of organ is not homogeneous within itself but is re-scaled with all the other maps in terms of photons per seconds acquire by the detector. Then the graphical representation rescaled the colour code as more homogeneously as possible to give the possibility to compare the different groups of maps as much as possible (this is clearly written in the text in line 403)

Request for change in the manuscript:

Line 403 only states: … By rescaling the map, it seems that Si is present quite uniformly inside the fruit … The mere mention of “rescaling” is completely inadequate to me. Again, like in many other responses by the authors, they explain things to me, the reviewer, reasonably well but fail to transfer this into the manuscript.

In my view, the explanation given in the author response should be included in the methods segment “Section 4.14 - Low Energy µ-XRF”, probably towards the end.

A possible description could be related to the response text above (or something similar the authors may compose):

The concentration maps shown in Figures 5 to 9 were generated in the following way: each group of maps for every treatment / type of organ is not homogeneous within itself but needs to be re-scaled with all the other maps in terms of photons per seconds acquire by the detector. Then the graphical representation rescaled the colour code as homogeneously as possible to give the possibility to compare the different groups of maps as much as possible. (Here I merely transposed the above response text – adequate formulation needs to be implemented by the authors).

Answe3

We thank the reviewer for helping us improving the manuscript, making it clearer for the readers. Indeed, as suggested, we inserted a sentence at the end of the methods segment “Section 4.14 - Low Energy µ-XRF”.

Lines:900-905 “The elemental maps shown in Figures 5 to 9 were generated in the following way: each group of maps for every treatment / type of organ is not homogeneous in terms of intensity counts within itself thus it needs to be rescaled with all the other maps in terms of photons per seconds acquired by the detector. Then in the graphical representation the colour code was rescaled as homogeneously as possible to give the possibility to compare the different groups of maps as much as possible.”

Questions a), b), c), d) 

(3)    In the Supplement the following minor issues need to be addressed:

(a)    Figure S4b – there is no vertical axis scale for the Gladis Si low plot

(b)    Figure S5a – the x-axis annotation for Aragon leaf is different (horizontal, not 45°)

(c)    Figure S5b – the x-axis annotation for Gladis fruit is different (horizontal, not 45°)

Answers a), b), c), d) 

We thank the reviewer for the observations.

We have provided to correct the figures and to homogenize the annotations angles on the x axis making them all parallel to the x axis.

Question d)

(d) All captions for Figures S4a / S4b / S5a / S5b: The difference between the annotation with ** (2-star) and *** (3-star) needs to be explained. The caption only states:  … The asterisks near the elements indicate those that have an interaction between treatment and part of the plant.

Answer d)

We thank the reviewer for the observation.  In Figures S4a / S4b / S5a / S5b the asterisks have been removed and a clearer Table with the two-way ANOVA the F statistics and the p values for treatment and plant part and their interactions, has been inserted in the text as table 1. See answer to Question 2.

Reviewer 2 Report (Previous Reviewer 2)

In this revised version I have noticed that authors present additional informations that enriched the discussion section and made improvements throughtout the manuscript. In this sense the authors responded my initial concerns.

Author Response

We greatly appreciate the acknowledgment that reviewer 2 gave on our work to improve the manuscript.

Reviewer 3 Report (Previous Reviewer 3)

Accepted

Author Response

Answer to Reviewer 3

Thank you for accepting the manuscript

This manuscript is a resubmission of an earlier submission. The following is a list of the peer review reports and author responses from that submission.

Round 1

Reviewer 1 Report

In the first instance, I wish to state that I am not an expert in the topic covered in this manuscript, so some of my perceptions might not be fully correct. Notwithstanding this, I think that the manuscript in its current form is not suitable for publication.

Below I list my observations and concerns, which lead me to my decision of rejection.

(1) Already the introduction suffers severely from a focus with respect to the topic of investigation –which continues in the main text. In particular, in my opinion, the authors did not address (i) the relevant details of the study, with respect to tomatoes; and (ii) what its actual aims were.
This lack of focus continues in the introduction, where the text meanders along with quite a few paragraphs, which in my view are irrelevant for the understanding of the research described later on. For example, already the second paragraph is questionable – it does not bear at all on the
content of the study who and when and how many tons of tomatoes were produced. Also, the list of (element) uptake by many different species other than tomato (lines 75-100) is excessively long, in my view even superfluous – the authors should concentrate on the stated topic:
tomatoes. Equally, the plant classification (lines 116-121) is irrelevant in the context of tomatoes as well. The whole of the Introduction would need restructuring, to highlight the aims of the study and its relevance.

(2) Right at the beginning of the section “Results” Figures 1 & 2 fall quite out of the blue, without any brief explanation which physiological and chemical analysis aspects they actually address, what the meaning of the discriminant functions 1 and 2 in the context of the analysis is (in the
Supplement similar plots are shown which exhibit completely different scatter); and what preliminary conclusion one can draw from them.
To me it confirms my experience that multivariate analysis tools are all too often used like black boxes, without clearly understanding internal mechanisms. Of course, one does not need to delve into any nitty gritty detail, but what input is actually used by the software, and how one
should interpret the output. In my view, none of this becomes clear, neither from the figures, nor from the associated text.

(3) Table 1 is – in my view – not relevant for the main text. If included at all it should be moved to the Supplement. Furthermore, I am at a complete loss as to the meaning of the letter combinations: “... lowercase letters indicate the statistical significance within treatments, uppercase letters indicate statistical significance within parts of the plant. Different letters indicate significantly different values p<0.005 ...” The whole table is like gobble-di-gook to me.

(4) Segment 2.2.2 leaves me at a complete loss, why one needs the various components (a) to (i) collated in Figure 3 – even in the Section “Methods” their relevance is not explained. Maybe I am ignorant, but I think that some explanation of parameter meaning belongs into any scientific
study. In addition, I think that some sort of preliminary conclusion about the overall relevance of the individual contributions for any later judgement of how the three tomato samples behave should be included in this section – as far as I can make out there’s no further discussion on this later in the manuscript.

(5) The same ragged description continues in the following Segments 2.2.3 to 2.2.6 – they consist mostly of a tiring list of reagents / reagent states, without much explanation as to their relevance in the context of this tomato study. All Segments would need to be made more comprehensible,
and highlight relevance for the study.

(6) I have similar concerns for Section 2.3 – Chemical analysis. Again, there is a long list (of individual elements), without any concernable explanation of their relevance.

(7) And then comes Section 2.4 in which the authors present their results from mXRT. It constitutes a procession of images in Figures 5-9, without much in terms of explanation. Furthermore, only images for the control and high-Si specimen are shown but none for low-Si – why?. Again, I
have to come back to my general criticism voiced quite a few times already: the study is supposed to highlight the importance of silicon but in lengthdeviates into other elements and molecules. If the authors thought that the other components were useful to display as images,
other than mentioning their role in the text, such plots should have been confined to the Supplement. In the end I asked myself: what is the point of all this? Proper, coherent discussion of relevance is mostly missing, in my view: ... is the aim to demonstrate differences / similarities in Si-accumulation in various parts of the plant? ... is it related to plant resilience or tomato yield? I can’t make out a final conclusion which is linked to a stated goal.

(8) Finally, while the various aspects of sample preparation and methodologies used look fine to me, once more I am lost as to the relevance and need for various analyses: which ones is important and which others are complementary / nice to include. As it stands, to me it only looks like a list of what the authors have as tools available to them and have then used.

(9) One point of organisation: the reference list breaks the numbering after reference 106 (107 is missing) and then continues with different numbering which initially confused me – only realising later that said second part refers to the references in the Supplement, where actually the numbering continues from 108 onwards.

Author Response

REVIEWER 1

Question 1

In the first instance, I wish to state that I am not an expert in the topic covered in this manuscript, so some of my perceptions might not be fully correct. Notwithstanding this, I think that the manuscript in its current form is not suitable for publication.

Below I list my observations and concerns, which lead me to my decision of rejection.

(1) Already the introduction suffers severely from a focus with respect to the topic of investigation –which continues in the main text. In particular, in my opinion, the authors did not address (i) the relevant details of the study, with respect to tomatoes; and (ii) what its actual aims were.
This lack of focus continues in the introduction, where the text meanders along with quite a few paragraphs, which in my view are irrelevant for the understanding of the research described later on. For example, already the second paragraph is questionable – it does not bear at all on the
content of the study who and when and how many tons of tomatoes were produced. Also, the list of (element) uptake by many different species other than tomato (lines 75-100) is excessively long, in my view even superfluous – the authors should concentrate on the stated topic:
tomatoes. Equally, the plant classification (lines 116-121) is irrelevant in the context of tomatoes as well. The whole of the Introduction would need restructuring, to highlight the aims of the study and its relevance.

Answer 1

We thank the reviewer for these important observations. The introduction has been reduced as much as possible, but we have to consider that also some explanations on silicon are warranted at the beginning of the manuscript.

Question2
(2) Right at the beginning of the section “Results” Figures 1 & 2 fall quite out of the blue, without any brief explanation which physiological and chemical analysis aspects they actually address, what the meaning of the discriminant functions 1 and 2 in the context of the analysis is (in the
Supplement similar plots are shown which exhibit completely different scatter); and what preliminary conclusion one can draw from them.
To me it confirms my experience that multivariate analysis tools are all too often used like black boxes, without clearly understanding internal mechanisms. Of course, one does not need to delve into any nitty gritty detail, but what input is actually used by the software, and how one
should interpret the output. In my view, none of this becomes clear, neither from the figures, nor from the associated text.

Answer 2

We thank reviewer 2 for this comment. The Canonical Discriminant Function Analysis utilizes the two Canonical Functions that result from the Factor reduction of the groups of independent variables (three in our case: Control, Si Low, Si High) to create a plot where the first canonical Function is on the x axis and tells us how much variability exists among groups of dependent variables and the second Canonical function is on the y axis and tells us how much variability exists within the each group of independent functions. Thus, if for example the groups are tightly packed together, but far apart it means that there is a great difference among the different groups of variables, but not much difference within the groups of the independent variables themselves. It is the Opposite of a MANOVA (Multivariate Analysis of Variance) which looks at whether groups differ along a linear combination of outcomes variables, discriminant analysis, unlike univariate Fs, breaks down or “discriminate” a set of groups using several predictors for example in the case of figure S1 the physiological Parameter, in case of Figure1 the treatments, in case of Figure 2 the parts of the plant. (A. Field, Discovering Statistics using IBM SPSS Statistics 5th Edition,2018, SAGE publication, 1 Oliver’s Yard, 55 city Road, London EC1Y1SP, UK).

We have inserted the following explanation in the text at lines 155-160:

“The Canonical Discriminant Function Analysis utilizes the two Canonical Functions that result from the Factor reduction of the groups of independent variables (three in our case: Control, Si Low, Si High) to create a plot where the first canonical Function is on the x axis and tells us how much variability exists among groups of dependent variables and the second Canonical function is on the y axis and tells us how much variability exists within the each group of independent functions.”

Question3
(3) Table 1 is – in my view – not relevant for the main text. If included at all it should be moved to the Supplement. Furthermore, I am at a complete loss as to the meaning of the letter combinations: “... lowercase letters indicate the statistical significance within treatments, uppercase letters indicate statistical significance within parts of the plant. Different letters indicate significantly different values p<0.005 ...” The whole table is like gobble-di-gook to me.

Answer3

We thank the reviewer for this insight in Table1. We move it to the supplementary materials where is now Table S 3.

Unfortunately, it cannot be changed in its nature because it is the combination of the variability among groups and within groups. The alternative would have been two identical tables one with the statistics for the variability among the parts of the plants and another for the variability among treatments. It had also the advantage to point out the elements (first row) for which there was interaction between treatment and plant part, which is important for assessing the effect of Si on the plant’s elemental homeostasis.

Question4
(4) Segment 2.2.2 leaves me at a complete loss, why one needs the various components (a) to (i) collated in Figure 3 – even in the Section “Methods” their relevance is not explained. Maybe I am ignorant, but I think that some explanation of parameter meaning belongs into any scientific
study. In addition, I think that some sort of preliminary conclusion about the overall relevance of the individual contributions for any later judgement of how the three tomato samples behave should be included in this section – as far as I can make out there’s no further discussion on this later in the manuscript.

Answer 4

We thank the reviewer for the comments. We have added more comments on this results in the discussion part. See lines 546-558:

“In fact H2O2 did not diminish with the increase of Si treatment in any circumstances (Fig 3b and 4b). This trend is followed also by the non-enzymatic defence such as the production of proline, which is accumulated as a common physiological response to various stresses, osmotic and oxidative [49, 63] or of Phenols, which serves as protectors against ROS as they are ROS chaperones [64, 65] (Figure 3a, 4 a and d).Also the malondialdehyde (MDA), which gives a measure of the lipid oxidation by the ROS, did not follow a statistical significant trend different from treatment to treatment, but only different within the plant parts The trend followed by all the antioxidant enzymes  was strictly cultivar dependent”.

Question5
(5) The same ragged description continues in the following Segments 2.2.3 to 2.2.6 – they consist mostly of a tiring list of reagents / reagent states, without much explanation as to their relevance in the context of this tomato study. All Segments would need to be made more comprehensible,
and highlight relevance for the study.

Answer 5

We thank the reviewer 2 for the suggestions. At the end of section 2.2.3 we have added:

” These parameters are all relevant to this study because they provide the state of the oxidative balance within the cells of the different plant organs which is the scope of Si treatments. In conditions of stress, we would expect these parameters to follow different trends from those described above were the antioxidant enzyme increase to counter the attack of ROS to the various cell parts”

At the end of section 2.2.4 we have clarified with the sentence:” Interestingly this can support the idea that Si might be adhered to the root”.

In the middle of paragraph 2.2.5 we have added at Lines 264-268:” Proline usually increased in all the plant parts in condition of osmotic and oxidative stress because it has the double function to promote water retention and turgor and to quench ROS. Here it is important to notice that the proline trend does not follow the Silicon concentrations treatments as it is found usually in condition of stress [63].”

At the end of the same paragraph, we have added the sentence:”). As we have observed in the cultivar Aragon there was not a trend followed by the antioxidant enzymes that points to the homogeneous response to a stress, but rather an increase and decrease linked to the plant parts.”

The section on results did not abound in such type of explanation because we thought they belonged to the Discussion section, where we compare the two cultivars regarding their overall response to the treatments.

Question6

(6) I have similar concerns for Section 2.3 – Chemical analysis. Again, there is a long list (of individual elements), without any concernable explanation of their relevance.

Answer 6

In this part we did not include any comment because we thought they pertain to the discussion. Therefore, to avoid useless repetitions, for both cultivars, we left all types of comments to the Discussion.

Question 7

(7) And then comes Section 2.4 in which the authors present their results from mXRT. It constitutes a procession of images in Figures 5-9, without much in terms of explanation. Furthermore, only images for the control and high-Si specimen are shown but none for low-Si – why?. Again, I
have to come back to my general criticism voiced quite a few times already: the study is supposed to highlight the importance of silicon but in length deviates into other elements and molecules. If the authors thought that the other components were useful to display as images,
other than mentioning their role in the text, such plots should have been confined to the Supplement. In the end I asked myself: what is the point of all this? Proper, coherent discussion of relevance is mostly missing, in my view: ... is the aim to demonstrate differences / similarities in Si-accumulation in various parts of the plant? ... is it related to plant resilience or tomato yield? I can’t make out a final conclusion which is linked to a stated goal.

Answer 7

The part of text in the paragraph 2.4 has been integrated at lines 371-376:

“The behaviors of Si have been inserted because they are also important for the homeostasis of the plants and to understand the mechanisms after Si treatments. Except for Aluminum, their distribution allows to better visualise the sample morphology as they are endogenous elements There are no maps for the treatments Si Low because we did not have enough beamtime at the synchrotron to perform also those analyses.”

We left to the discussion the interpretation of the images in the context of this study which is to highlight the cellular distribution of Silicon in primis, but also of other minerals of interest because important for the cell metabolism and because they are certainly part of the mechanisms of Si response.

The important of Si is related also to how its application to the plant influences the homeostasis of other minerals. In fact, we found that after the application of different Si concentrations, As as shown in Table 1, now Table S2, modifies the concentrations of many other elements that are important for the cell and the plant metabolism, such as Mg, K, Mn, Cu, Zn, in all the cultivars and in others depending on the cultivar.

The meaning of all this work is to study the mechanisms after the treatment of tomato plant with Si when the plant is not under stress, which has never done before. It is related both to the plant resilience to different silicon treatments and to the possible change in yield that these treatments can elicit. We have not found any significant change in yield for any of the Si treatment in comparison to the control (TableS1), but both cultivars are resilient to all the Si treatments in comparison to the control.  The final conclusions that we drew from all this work were that the response of the plants, when not under stress, to Silicon treatments at different concentrations is very different from the response when the plant is under any type of biotic or abiotic stresses. In those cases, the concentration of Si it matters because usually at higher concentration the plant response is generally better than at low concentrations. Moreover, the level of enzymatic, and non-enzymatic antioxidants followed trend that is to increase or diminish according to their function in quenching the ROS produced by the stress. What emerges here is a complex picture which ask for more in deep molecular and physiological analysis through the data present here are already well significant. We added at the end of the Discussion the following sentence: Lines: 637-639

“We must admit however that the topic is still not completely elucidated in several points and warrants further investigation.”

Question8
(8) Finally, while the various aspects of sample preparation and methodologies used look fine to me, once more I am lost as to the relevance and need for various analyses: which ones is important and which others are complementary / nice to include. As it stands, to me it only looks like a list of what the authors have as tools available to them and have then used.

Answer 8

We thank the reviewer for the comment. We wish to convey the message that none of our analysis is complementary. All the physiological enzymatic analyses and non-enzymatic analyses were instrumental to assess the plant redox state due to the treatments with Silicon; the chemical analyses were important to determine how Si influences the plant element homeostasis; the molecular analyses had the scope to highlight possible genes markers of Si treatments. We described the method as clearly as possible so that anybody willing to reproduce them has the possibility to follow precise instructions.

Question9
(9) One point of organisation: the reference list breaks the numbering after reference 106 (107 is missing) and then continues with different numbering which initially confused me – only realising later that said second part refers to the references in the Supplement, where actually the numbering continues from 108 onwards.

Answer 9

The numbers changes were requested by the Editor.

Reviewer 2 Report

The manuscript describes the responses of two cultivars of tomato (Aragon and Gladis) under two different treatments of Si. The authors used a combination of morphological, biochemical, and molecular approach to study the effects of Si in tomato. The authors suggested that some treatment with Si might have a priming effect on the plant according to their responsiveness to Si. In general, the manuscript is well-structured and written. I have just some considerations that are described below.

“Abstract”

I think that the abstract should be clearer and show the reader an overview of the study. Thus, the authors could improve the abstract highlighting better the purpose of the research.

“Results”

Page 4. The quality of figure 1 and 2 could be improved by increasing the font size and the colors of the dots in the image.

Page 5, What mean " We cannot exclude here a technical mistake". Please explain it, and exclude this statement.

Page 11. Figure 5. XRF (micro-X-ray Fluorescence) for element distribution. Are the figures in the same scale bar? The elemental maps of the roots of the tomato cultivars are at different scales. Please standardize.

Regarding the analysis of transcriptional expression: what was the choice criteria for selection of target genes?

The results show that the treatments repressed gene expression, particularly genes related to ROS production. So, what explains the production of high levels of hydrogen peroxide in tomato cultivars after Si treatments?

Page 1, Figure 10. Still regarding the expression analysis, the transcriptional analysis of the target genes in the non-treated (NT) samples were not performed? This group should be included in the gene expression analysis.

Minor

Please standardize "Si low or Si Low, Si high or Si High”

Page 17, “Figure 10” Figure 10.

Author Response

REVIEWER 2

The manuscript describes the responses of two cultivars of tomato (Aragon and Gladis) under two different treatments of Si. The authors used a combination of morphological, biochemical, and molecular approach to study the effects of Si in tomato. The authors suggested that some treatment with Si might have a priming effect on the plant according to their responsiveness to Si. In general, the manuscript is well-structured and written. I have just some considerations that are described below.

Question 1

“Abstract”

I think that the abstract should be clearer and show the reader an overview of the study. Thus, the authors could improve the abstract highlighting better the purpose of the research.

Answer 1

We thank the reviewer for the observation. We have inserted in the abstract the sentence:

“The scope of the study was to highlight any significant response of the plants to the Si treatments, trying to find similarities with any response to Si of plants under stress.”

Question 2

“Results”

Page 4. The quality of figure 1 and 2 could be improved by increasing the font size and the colors of the dots in the image.

Page 5, What mean " We cannot exclude here a technical mistake". Please explain it and exclude this statement.

Answer 2

We thank the reviewer for the remarks. We have improved the quality of figure 1 and 2 in the text and also of Figure S1 and Figure S2 in the Supplementary by increasing the font size, the colours and size of the scatterplots as much as SPSS allowed us to do.

The sentence in Page 5 referred to a possible instrumental error. It has been eliminated.

Question 3

 Page 11. Figure 5. XRF (micro-X-ray Fluorescence) for element distribution. Are the figures in the same scale bar? The elemental maps of the roots of the tomato cultivars are at different scales. Please standardize.

Answer 3

We thank the reviewer for this remark. Indeed, it would be better to put the images on the same scale but since the scale is very different, esthetically the figure wouldn’t be very nice. We attach it here below. We prefer to keep the old version of the figure. However, if the reviewer insists we can use the following one:

Question 4

Regarding the analysis of transcriptional expression: what was the choice criteria for selection of target genes?

The results show that the treatments repressed gene expression, particularly genes related to ROS production. So, what explains the production of high levels of hydrogen peroxide in tomato cultivars after Si treatments?

Answer 4

We thank the reviewer for this important observation. The only gene that is related to the ROS activity such as that of H2O2 is the Cytochrome c oxidase 1 COX and is mostly upregulated by the Si treatments in both cultivars.

We have added an explanation for this in the Discussion in the text at Lines 552-557:

“In fact, H2O2 did not diminish with the increase of Si treatment in any circumstances (Fig 3b and 4b), On the contrary it increases, in fact COX1, GAPDH and CAO the three genes mostly devoted to increasing ROS activity were downregulated by the treatments. (Figure 10). Clearly the treatments with Si must have activated other genes for the production of ROS in particular H2O2.that we have not taken into consideration.”

Question 4

Page 1, Figure 10. Still regarding the expression analysis, the transcriptional analysis of the target genes in the non-treated (NT) samples were not performed? This group should be included in the gene expression analysis.

Answer 4

The transcriptional analysis of the target genes in the non-treated (NT) samples were performed and used to normalized data: the ΔCt of Nt sample was subtracted from the ΔCt of treated samples. Then the “log2 fold change” was calculated and represented in figure 10.

We added in the Figure Caption the explanation:

“Data were normalized on NT samples”

Minor

Please standardize "Si low or Si Low, Si high or Si High”

We thank the reviewer for this observation. We have standardiside the names of the treatments.

Page 17, “Figure 10” Figure 10.

We thank the reviewer for this observation. The error has been corrected

REVIEWER 2

The manuscript describes the responses of two cultivars of tomato (Aragon and Gladis) under two different treatments of Si. The authors used a combination of morphological, biochemical, and molecular approach to study the effects of Si in tomato. The authors suggested that some treatment with Si might have a priming effect on the plant according to their responsiveness to Si. In general, the manuscript is well-structured and written. I have just some considerations that are described below.

Question 1

“Abstract”

I think that the abstract should be clearer and show the reader an overview of the study. Thus, the authors could improve the abstract highlighting better the purpose of the research.

Answer 1

We thank the reviewer for the observation. We have inserted in the abstract the sentence:

“The scope of the study was to highlight any significant response of the plants to the Si treatments, trying to find similarities with any response to Si of plants under stress.”

Question 2

“Results”

Page 4. The quality of figure 1 and 2 could be improved by increasing the font size and the colors of the dots in the image.

Page 5, What mean " We cannot exclude here a technical mistake". Please explain it and exclude this statement.

Answer 2

We thank the reviewer for the remarks. We have improved the quality of figure 1 and 2 in the text and also of Figure S1 and Figure S2 in the Supplementary by increasing the font size, the colours and size of the scatterplots as much as SPSS allowed us to do.

The sentence in Page 5 referred to a possible instrumental error. It has been eliminated.

Question 3

 Page 11. Figure 5. XRF (micro-X-ray Fluorescence) for element distribution. Are the figures in the same scale bar? The elemental maps of the roots of the tomato cultivars are at different scales. Please standardize.

Answer 3

We thank the reviewer for this remark. Indeed, it would be better to put the images on the same scale but since the scale is very different, esthetically the figure wouldn’t be very nice. We attach it here below. We prefer to keep the old version of the figure. However, if the reviewer insists we can use the following one:

Question 4

Regarding the analysis of transcriptional expression: what was the choice criteria for selection of target genes?

The results show that the treatments repressed gene expression, particularly genes related to ROS production. So, what explains the production of high levels of hydrogen peroxide in tomato cultivars after Si treatments?

Answer 4

We thank the reviewer for this important observation. The only gene that is related to the ROS activity such as that of H2O2 is the Cytochrome c oxidase 1 COX and is mostly upregulated by the Si treatments in both cultivars.

We have added an explanation for this in the Discussion in the text at Lines 552-557:

“In fact, H2O2 did not diminish with the increase of Si treatment in any circumstances (Fig 3b and 4b), On the contrary it increases, in fact COX1, GAPDH and CAO the three genes mostly devoted to increasing ROS activity were downregulated by the treatments. (Figure 10). Clearly the treatments with Si must have activated other genes for the production of ROS in particular H2O2.that we have not taken into consideration.”

Question 4

Page 1, Figure 10. Still regarding the expression analysis, the transcriptional analysis of the target genes in the non-treated (NT) samples were not performed? This group should be included in the gene expression analysis.

Answer 4

The transcriptional analysis of the target genes in the non-treated (NT) samples were performed and used to normalized data: the ΔCt of Nt sample was subtracted from the ΔCt of treated samples. Then the “log2 fold change” was calculated and represented in figure 10.

We added in the Figure Caption the explanation:

“Data were normalized on NT samples”

Minor

Please standardize "Si low or Si Low, Si high or Si High”

We thank the reviewer for this observation. We have standardiside the names of the treatments.

Page 17, “Figure 10” Figure 10.

We thank the reviewer for this observation. The error has been corrected

Reviewer 3 Report

In abstract, conclusion section rewrite from 26 to 27.. clearly rewrite the findings in conclusion.

Keywords, remove typo "Silicon" s small not capital.

Line 117, remove typo 2002 remove,

From line 31 to 127; small paragraph combined into one paragraph.

Line 51; "As" First time use full form.

The decimal of value reported in all tables of the manuscript must be uniform as the value of the detection limit of the used method.

Figure quality very low,,, improve figure resolution.

Heading 2.2.3, line 5, use abberviation of H2O2,

very difficult to review without line number,,,

resubmit manuscript after revising above comments, i will revise the manuscript again,,

half of the manuscript have line number and half not,,, 

Author Response

REVIEWER 3

Question 1

In abstract, conclusion section rewrite from 26 to 27.. clearly rewrite the findings in conclusion.

Answer 1

We thank the reviewer for the comment. We rewrote the abstract thus:

“The work focussed on the analysis of two cultivars of tomato (Solanum lycopersicum L.), Aragon and Gladis, under two different treatments of silicon, Low, 2 L of 0.1 mM CaSiO3, and High, 0.5 mM CaSiO3, weekly, for 8 weeks, under stress free conditions. We subsequently analysed the morphology, chemical composition and elemental distribution using synchrotron-based µ-XRF techniques, physiological, and molecular aspects of the response of the two cultivars. The scope of the study was to highlight any significant response of the plants to the Si treatments, trying to find similarities with any response to Si of plants under stress. The results demonstrated that the response was mainly cultivar dependent, and that it did not differ from the two types of treatments. Si deposited mainly in the cell walls of the cells of fruits, leaves, and roots the treatments did not elicit many significant changes from the point of view of the total elemental content, the physiological parameters that measured the oxidative stress, and the transcriptomic analyses focalised on genes related to the response to Si. We observed a priming effect of the treatment on the most responsive cultivar, Aragon, in respect to future stress, while in Gladis the Si treatment did not change significantly the measured parameters”

Question2
Keywords, remove typo "Silicon" s small not capital.

Answer 2

The typo has been corrected

Question3
Line 117, remove typo 2002 remove,

Answer 3

The typo has been removed.

Question 4

From line 31 to 127; small paragraph combined into one paragraph.

Answer 4

We have made one single paragraph from lines 31 to 130

Question 5
Line 51; "As" First time use full form.

Answer 5

Question 6

The decimal of value reported in all tables of the manuscript must be uniform as the value of the detection limit of the used method.

Answer 6

We have corrected the tables.

Question 7

Figure quality very low,,, improve figure resolution.

Answer 7

We have tried to improve the images as much as we could.

Question 8

Heading 2.2.3, line 5, use abberviation of H2O2,

Answer 8

The abbreviation has been inserted instead of the name

Question 9

very difficult to review without line number,,,

Answer 9

The lines numbers are all on the right side from top to bottom in the entire manuscript. If you received a version without lines numbers is due to the passage from doc.x to the  pdf version done by the journal system. Sorry about that it was not our fault.

Questio 10

resubmit manuscript after revising above comments, i will revise the manuscript again,,

half of the manuscript have line number and half not,,, 

Answer 10

We thank the reviewer for this comment. Let us reiterate that The lines numbers are all on the right side from top to bottom in the entire manuscript. If you received a version without lines numbers is due to the passage from doc.x to pdf done by the Journal system. Sorry about that, it was not our fault.

Round 2

Reviewer 1 Report

As a whole, the text has improved quite a bit, where the authors have amended the narrative according to the suggestions provided in the first round of reviews.

However, other issues raised by the reviewer(s) have been to a certain degree answered in the authors’ response letter but didn’t find their way into the manuscript – that defeats the object, because it is the general reader – and not only the reviewer – who need to benefit from clarifications.

Having said that, I still think that the manuscript in its current form is

     not yet ready for publication, but requires further, SUBSTANTIAL revision.

In particular, in my view it is – by and large – the statistical data analysis, data presentation and conclusions drawn from tables, graphs and images, which need major improvement. Some of the related issues were already noted in the first round, but not adhered to (e.g. simply shifting the former Table 1 to the Supplement in form of Table 3 – without addressing the queries made with respect to the original table.

Because of this table transfer I also scrutinised the other Supplement entries further; many of those require substantial improvement, better link to the main manuscript text and proper interpretation. As they stand at the moment, they do not really represent “added value”.

Here are my concerns regarding the revised manuscript:

Figures 3 & 4

In the two figures a number of “letters”, and combinations thereof accompany, the bar graph data: a, b, c, d, e and f as well as ab, ac, ad, ae, af, bc, cb. What is the meaning of these? Are they outputs from the analysis software? Are they a personal annotation chosen by the authors? In either case, a proper explanation has to be given and each letter associated with a numerical value (if there is any), or a particular (statistical) meaning. Otherwise, it is next to impossible for a non-expert reader to interpret the figures and gauge any detailed significance related to the study.

mXRF – Section 2.4 and Figures 5-9

The introduction has become a bit clearer in the revised version of the manuscript. However, there are two diametrically opposing explanations as to why no mXRF maps are included for the low-Si treatment of the plants.

The first one [lines 343-344] – not sufficient beam time – to me is a quite valid one; if one has to rely on external equipment / facilities, time constraints may hamper any planned campaign. The second one ([lines 344-347] I find rather questionable. In my view, the Si-concentration data provided in Table S3a,S3b do not support the statement; those values are all the same within the large uncertainty of ~20% - to hinge the conclusion on one value being marginally larger than the others (Aragon Si-high), but still well within the uncertainty band, is statistically inconclusive.

Then in the text, various claims on “trends” extracted from the five figures are mentioned / discussed in Sections 2.4.1 and 2.4.2.

To come to such trend conclusions from the comparison of only two values / distribution maps (with the third missing due to the non-availability of equipment), is rather bold in my view. If one looks at the related numerical values of total element concentrations, collated in Table S3a,S3b, then I can’t really see some of the aforementioned trends.

Specifically, I have great difficulty in verifying the claimed relations respective “enhanced at the cell walls / distributed throughout the cell”. Some of the interpreted trends could be mere artefacts of the false-colour representation of spatial concentration differences.

The dynamic amplitude ranges are substantially different in many cases but carry the same colour-coding from minimum to maximum; this can severely distort the perception about the actual semi-quantitative distribution of e.g. a specific element – different colours can be associated with the same concentration, and therefore give the false impression that something has changed in two differently-scaled images. This issue needs to be addressed very carefully before significant conclusions may be drawn.

Table S2 plus related statements in the main text

[line 121] – the table is mentioned but with little further detail

[lines 489-493] – the table is mentioned, with a few words on plant yield but without much of discussion

[lines 645-653] – information on sampling is given, but without any details on sampling sizes (I assume 1 for each plant for the roots and the fruits (ripening-dependent), but how many actually for the aerial (leaf) part? With respect to the latter, since I assume that more than one leaf per plant was measured, were the statistics done for the whole set, or for each plant individually and then averaged at the end?

The actual deduction of conclusions may strongly depend on the sample size (see e.g. the brief summary given by CJ Morgan, 2017, on the issue). I think with the in general small sample sizes in this study, neither the shape of the error distribution nor the presence of outliers can be judged with great confidence. Therefore, at least one should look at extended descriptive statistics (i.e. mean, median, mode, range, and quartile deviation), rather than only the mean and the standard deviation.

Table S2 itself stands a bit like an orphan in the Supplement, without any brief explanatory summary of the (few) scattered remarks on it in the main text. Without some words on sample sizes, statistics used (the authors mainly mention the use of ANOVA in the main text) and some interpretation on statistical relevance, to me the table does not really sparkle as to what relevance it carries for the interpretation of results – mostly it is only hand-waving mention in the main text.

   Reference

Morgan, C.J. Use of proper statistical techniques for research studies with small samples. Am J Physiol Lung Cell Mol Physiol 2017, 313, L873-L877. https://doi.org/0.1152/ajplung.00238.2017

Tables S3a & S3b plus related statements in the main text

[line 281]:

2.3.1. Elements in Cv. Aragon.  --  Table S3a reports the two-way ANOVA, in the first column it is possible to observe the interaction between treatments and plant parts 

?????   I don’t understand what this is supposed to mean – in the 1st column the elements are listed

[line 306]:

2.3.2. Elements in Cv. Gladis.  --  The two-way ANOVA in Table S3b evidenced interactions between plant parts and treatments  The meaning of ANOVA is only given much later in the manuscript; it should be given the first time it appears in the text – not everybody may be immediately familiar with it.

Letter combinations in the table columns: aA / aB / aC / bA / bB / bC. For what do these actually stand? Do they stem from the output of the analysis software, or are they introduced by the authors, so they  themselves quickly can compare statistical relevance?  

I now get the link to lower case / upper case letters from the Table caption: … lowercase letters indicate the statistical significance within treatments, uppercase letters indicate statistical significance within parts of the plant

But with which numerical value are the letter a,b,c and A,B,C associated with? The caption states … different letters indicate significantly different values p<0.005 …, which to me doesn’t make much sense, since it is only a single value but there are three different entry types (letters).

And I am completely at a loss what the double entries in Table S3(a) actually mean: for example, (i) for the “root”-entries for Ni, these are aA & B; (ii) for some other elements different letters appear; and (iii) at times only one or two of the treatments show such a double entry. What on earth does that all mean? Nowhere is an explanation given. I mentioned this already in my first review, but still no explanation has been given

Finally, having struggled significantly to plough through the vast number of tabulated numerical values and trying to link them to the narrative in the text, I think the presentation should be made more transparent. My advice would be: display the data in bar-graph form, as was done for the molecular species in Figures 3 & 4. That would have made it much easier, at least for me, to visualise the results in a comparative manner – extracting trends from a maze of numbers is extremely more difficult, and thus blurs understanding, than from a concise graphical display.

Figure S1 – briefly mentioned in [line 124]

Figures S2/S3 – addressed in short sentences associated with their referencing in [lines 134, 483]

These tables are OK-ish to me. However, personally I think that a bit more explanation is required; the guesses are rather hand-waving, and given the wealth of measurements / data the authors have undertaken / accumulated, more could have been extracted (I can’t easily make out any major conclusion with respect to these figures in the discussion section). Also, the sentence preceding the referencing in line 483 is garbled: it simply doesn’t make sense to me.

Figure S4 – addressed in [lines 484-486]

Neither from the three lines of text, nor from the 3D display of PCA components in the figure, do I understand what one may learn from the analysis data and its display.

To me the elemental and physiological (molecular) data vectors look like a seemingly random scatter of end-points. There is no mention which actual data found their way into this analysis, and no explanation of any potential significance of the data distribution, other than that the clouds” seem to be at slightly different locations in the 3D plots.

What is the point? Either substantial explanation / discussion needs to be provided, or any link between certain parts in the discussion in the manuscript text be established (I can’t see any); or the figures should be eliminated – to simply show them because application of PCA was possible is futile.

Author Response

Dear Dr. Andreea Margareta Sinka

Assistant Editor, MDPI, Cluj,

MDPI Branch Office, Cluj, Romania

Dear Guest Editors

Dr. Alessandra Gianoncelli

ELETTRA Sincrotrone Trieste S.C.p.A., Basovizza, Italy

And

Dr. Lorella Pascolo

IRCCS Burlo Garofolo, Trieste, Italy

We are here resubmitting for the Special Issue "Spectroscopic Imaging Techniques for Biological Metabolisms Investigations” of the International Journal of Molecular Science a manuscript entitled: “Combination of Biochemical, Molecular, and synchrotron-radiation-based techniques to study the effects of Silicon in tomato (Solanum lycopersicum L.)” after the second round of revisions.

We have changed Tables S3a, b in graphs, now Figure S4a,b and Figure S5a,b as requested by reviewer 1 and eliminated the PCA and all the comments concerned with that as requested by reviewer 1. We have performed the non-parametric statistics on the data in Table 1. We have made all the possible changes requested by Reviewer 1 and answered to all the questions posed by that reviewer with changes in the text and images.

We hope the manuscript is now acceptable for publication.

Kind regards,

Dr. Marta Marmiroli                                                                                      Parma, 16/10/2022

Answers to Reviewer 1 – round 2

Reviewer 1

As a whole, the text has improved quite a bit, where the authors have amended the narrative according to the suggestions provided in the first round of reviews.

However, other issues raised by the reviewer(s) have been to a certain degree answered in the authors’ response letter but didn’t find their way into the manuscript – that defeats the object, because it is the general reader – and not only the reviewer – who need to benefit from clarifications.

Having said that, I still think that the manuscript in its current form is

     not yet ready for publication, but requires further, SUBSTANTIAL revision.

In particular, in my view it is – by and large – the statistical data analysis, data presentation and conclusions drawn from tables, graphs and images, which need major improvement. Some of the related issues were already noted in the first round, but not adhered to (e.g. simply shifting the former Table 1 to the Supplement in form of Table 3 – without addressing the queries made with respect to the original table.

Because of this table transfer I also scrutinised the other Supplement entries further; many of those require substantial improvement, better link to the main manuscript text and proper interpretation. As they stand at the moment, they do not really represent “added value”.

Here are my concerns regarding the revised manuscript:

Question 1 

Figures 3 & 4

In the two figures a number of “letters”, and combinations thereof accompany, the bar graph data: a, b, c, d, e and f as well as ab, ac, ad, ae, af, bc, cb. What is the meaning of these? Are they outputs from the analysis software? Are they a personal annotation chosen by the authors? In either case, a proper explanation has to be given and each letter associated with a numerical value (if there is any), or a particular (statistical) meaning. Otherwise, it is next to impossible for a non-expert reader to interpret the figures and gauge any detailed significance related to the study.

Answer 1

The multiple letters serve to indicate that a measure is between two values according to the Tukey HSD post hoc test that has been performed following the one-way Analysis of Variance (ANOVA) analysis of the data. The data on the roots leaf and fruit have been treated all together thus the reason for the presence of multiple letters. They can be considered as output from the SPSS software. The explanation is added in the Figure captions:

Caption Figure 3: “Effect of Si Low and Si High physiological parameters in cvs. Aragon: a) total phenolic (TP), b) hydrogen peroxide (H2O2), c) lipid peroxidation (MDA), d) ascorbate peroxidase (APX), e) proline, f) catalase activity (CAT), g) superoxide dismutase activity (SOD), h) peroxidase activity (POD), I) lipoxygenase (LOX), j) glutathione peroxidase (GPX), k) Glutathione S-transferases (GST). Different superscript letters above the standard deviation bars indicate significant differences according to ANOVA followed by post hoc Tukey’s HSD test for multiple comparisons analysis (p≤0.01). Values equal to 0 means below detection limit (nd). Multiple letters indicate that a value is statistically between the values corresponding to the two letters.”

Caption Figure 4: “Effect of Si Low and Si High physiological parameters in cvs. Gladis: a) total phenolic (TP), b) hydrogen peroxide (H2O2), c) lipid peroxidation (MDA), d) ascorbate peroxidase (APX), e) proline, f) catalase activity (CAT), g) superoxide dismutase activity (SOD), h) peroxidase activity (POD), I) lipoxygenase (LOX), j) glutathione peroxidase (GPX), k) Glutathione S-transferases (GST). Different superscript letters above the standard deviation bars indicate significant differences according to ANOVA followed by post hoc Tukey’s HSD test for multiple comparisons analysis (p≤0.01). Values equal to 0 means below detection limit (nd). Multiple letters indicate that a value is statistically between the values corresponding to the two letters.”

Question 2

 mXRF – Section 2.4 and Figures 5-9

The introduction has become a bit clearer in the revised version of the manuscript. However, there are two diametrically opposing explanations as to why no mXRF maps are included for the low-Si treatment of the plants.

The first one [lines 343-344] – not sufficient beam time – to me is a quite valid one; if one has to rely on external equipment / facilities, time constraints may hamper any planned campaign. The second one ([lines 344-347] I find rather questionable. In my view, the Si-concentration data provided in Table S3a,S3b do not support the statement; those values are all the same within the large uncertainty of ~20% - to hinge the conclusion on one value being marginally larger than the others (Aragon Si-high), but still well within the uncertainty band, is statistically inconclusive.

Then in the text, various claims on “trends” extracted from the five figures are mentioned / discussed in Sections 2.4.1 and 2.4.2.

To come to such trend conclusions from the comparison of only two values / distribution maps (with the third missing due to the non-availability of equipment), is rather bold in my view. If one looks at the related numerical values of total element concentrations, collated in Table S3a,S3b, then I can’t really see some of the aforementioned trends.

Specifically, I have great difficulty in verifying the claimed relations respective “enhanced at the cell walls / distributed throughout the cell”. Some of the interpreted trends could be mere artefacts of the false-colour representation of spatial concentration differences.

The dynamic amplitude ranges are substantially different in many cases but carry the same colour-coding from minimum to maximum; this can severely distort the perception about the actual semi-quantitative distribution of e.g. a specific element – different colours can be associated with the same concentration, and therefore give the false impression that something has changed in two differently-scaled images. This issue needs to be addressed very carefully before significant conclusions may be drawn.

 Answer 2

Regarding the lack of the measurements of the treatment Si Low, both explanations are true, but principally, we didn’t have enough beamtime.

 Regarding the false colour code, the explanation is that each group of maps for every treatment/ type of organ is not homogeneous within itself but is re-scaled with all the other maps in terms of photons per seconds acquire by the detector. Then the graphical representation rescaled the colour code as more homogeneously as possible to give the possibility to compare the different groups of maps as much as possible (this is clearly written in the text in line 403). This is the reason why we describe trends among single elements in different tissues and organs. In fact, we do not compare different elements among them because they have different intensities, which means different concentrations. In addition, do not forget that in these maps we are dealing with small parts of tissues thus the thickness can play an active role in modifying the measured concentrations of the elements. On the other hand, in the chemical analyses of tables S3a,b the measures are of the bulk content of the whole organ, thus they are measured in a fixed unit, which is ppm, and can be directly compared with a statistical approach. In any case any mention to “trends” have been eliminated from paragraph 2.4.1 and 2.4.2. The paragraphs are reported here below:

Lines 396-426: “2.4.1. Si distribution in Aragon

The distribution of Si within the cell root for the High treatment, is mainly on the cell wall. It is distributed in the parenchyma cells (Figure 5a), following the same pattern as Na and Mg. Aluminum is distributed inside the root in a uniform fashion within the cell, although slightly more on the cell wall (Figure 5a).

In fruit control (Figure 6a) there is a marked intense point of Si in a cell, co-localised with a hot spot of Al and with higher content of Mg, Na, O as well, thus it is likely a Si salt or on artefact of the preparation. By rescaling the map, it seems that Si is present quite uniformly inside the fruit. In Aragon fruit under the high treatment (Figure 6b) Si has a clearer pattern as it appears to be distributed inside the cell with higher concentrations in the center of the pericarp cell, and partially on the cell walls. The other elements are more intensely distributed in the cell walls. In the leaf control (Figure 7a), beside some hot spots, Si is mildly distributed in some cell walls, while in the leaf under the high treatment (Figure 7b), Si is clearly visible with Na, Mg, and Al in the cell walls.

2.4.2. Si distribution in Gladis

Silicon distribution in Gladis roots under the High treatment (Figure 5b), was mainly in the parenchyma and endodermis tissues, especially in the cell walls. Na and Mn follow the same distribution even if less concentrated. Aluminum is distributed mainly on the cell walls of the parenchyma and endodermis (Figure 5b). From the maps, Si content in treated Gladis roots appears to be lower than on Aragon treated roots (Figure 5a).

In Gladis control fruits (Figure 8a), Si seems distributed mainly along cell walls. It is possible to see that also the other elements are distributed in the cell walls of the pericarp. For the fruit treated with High concentrations of Si (Figure 8b), the element penetrated inside the cells, while Mg and Na remained in the cell wall. Al has a high concentration spot, corresponding to a high concentration spot in the Si map, which can correspond to an aluminosilicate aggregate. In the control of the leaf (Figure 9a), Si has a high intensity spot corresponding to the same position in the Al map, which could be an aluminosilicate aggregate. Beside this it appears to be distributed in the cell walls, co-localising with Mg and Na. For the map of the High Si treatment (Figure 9b), it is possible to confirm that Si is both within the leaf cells and on the cell walls, while Na, Al, and Mg are mainly on the cell walls”

Question 3

Table S2 plus related statements in the main text

[line 121] – the table is mentioned but with little further detail

[lines 489-493] – the table is mentioned, with a few words on plant yield but without much of discussion

Answer 3

At line 121-123 The following sentence have been added:” The means for the aerial parts are almost the same but with a higher standard errors in respect to the roots and the fruit.”

 And at lines 496-498 it has been added:”, in fact according to some literature the administration of Si increases the biomass, according to other it leaves it as it is, but it is all still cultivar dependent [49-51].”

Question 4

[lines 645-653] – information on sampling is given, but without any details on sampling sizes (I assume 1 for each plant for the roots and the fruits (ripening-dependent), but how many actually for the aerial (leaf) part? With respect to the latter, since I assume that more than one leaf per plant was measured, were the statistics done for the whole set, or for each plant individually and then averaged at the end?

Answer 4

We sampled the whole root which in the tomato consist of a principal root with several lateral roots. The fruits sampled were from three to five and the leave from three to four according to the cultivar the number varied. The statistics were averaged at the end and were done pooling all the plants together, three plants per.

Paragraph 4.4 has been rewritten thus:” 4.4 Sampling and morphological analysis

Samples were taken after 8 weeks of treatment.  The whole root system was collected (principal and lateral roots, washed in deionized water, length and weight were determined. Fruits were counted, weighed, and subdivided according to the ripening stage (green, breaker and red). Fruits at the same developmental stage, according to ‘days after flower anthesis and color’ and positioned between the 6th and 8th leaf nodes were selected for analysis, three to five per plant [8]. Fruits were washed in deionized water: the pericarp and cuticle were retained, while the placenta and seeds were discarded. The aerial part was measured for length and weight and middle leaflets of the same age and positioned between the 6th and 8th nodes up the stalk were collected for analysis, three to four according to the cultivar. Samples of the four specimens were group together before averaging. According to the analysis to be performed, fruits, leaves and roots were oven-dried or snap frozen in liquid nitrogen and stored at -80°C until use or assayed immediately after harvest.”

Question 5

The actual deduction of conclusions may strongly depend on the sample size (see e.g. the brief summary given by CJ Morgan, 2017, on the issue). I think with the in general small sample sizes in this study, neither the shape of the error distribution nor the presence of outliers can be judged with great confidence. Therefore, at least one should look at extended descriptive statistics (i.e. mean, median, mode, range, and quartile deviation), rather than only the mean and the standard deviation.

Table S2 itself stands a bit like an orphan in the Supplement, without any brief explanatory summary of the (few) scattered remarks on it in the main text. Without some words on sample sizes, statistics used (the authors mainly mention the use of ANOVA in the main text) and some interpretation on statistical relevance, to me the table does not really sparkle as to what relevance it carries for the interpretation of results – mostly it is only hand-waving mention in the main text.

   Reference

Morgan, C.J. Use of proper statistical techniques for research studies with small samples. Am J Physiol Lung Cell Mol Physiol 2017, 313, L873-L877. https://doi.org/0.1152/ajplung.00238.2017

Answer 5

After reading the paper by C.J. Morgan, the Authors decided to try to apply to the data in Table S2 on Morphological analyses the non-parametric statistics of the Kruskal -Wallis (one way-nonparametric ANOVA) to see if there were any significative differences.

The results are shown here below for the reviewer 1 perusal. Since there are no significative differences in any of the measurements for any of the treatments for both cultivars, we decided to keep table S2 as it is.

 Kruskal-Wallis test for independent samples for all the samples in Tables S2 grouped according to treatment for the cultivar Aragon.

 Kruskal-Wallis test for independent samples for all the samples in Tables S2 grouped according to treatment for the cultivar Gladis.

Question 6

Tables S3a & S3b plus related statements in the main text

 [line 281]:

2.3.1. Elements in Cv. Aragon.  --  Table S3a reports the two-way ANOVA, in the first column it is possible to observe the interaction between treatments and plant parts … 

?????   I don’t understand what this is supposed to mean – in the 1st column the elements are listed

Answer 6

When there is a two-way ANOVA there is always an interaction between the independent factors that is visible in the dependent factors. In the first column of table S3a,b the asterisks under the elements indicate if and how strong according to the number of asterisks, the interaction is for each element considering their variation in combination of the independent variables, which in this case are the treatments and the plant parts.

Question 7

[line 306]:

2.3.2. Elements in Cv. Gladis.  --  The two-way ANOVA in Table S3b evidenced interactions between plant parts and treatments …  The meaning of ANOVA is only given much later in the manuscript; it should be given the first time it appears in the text – not everybody may be immediately familiar with it.

Answer 7

The meaning of ANOVA is given in the section on statistics in the Materials and Methods, that is its proper place, if anybody has a doubt they just have to scroll down the text and  reach the correct section. It is not the Author’s fault if the Journal format puts the Materials and Methods as the last section in the manuscript. According to this reasoning also many other concept related to the experimental parts reported in the results should be explained earlier, but are not.

 Question 8

Letter combinations in the table columns: aA / aB / aC / bA / bB / bC. For what do these actually stand? Do they stem from the output of the analysis software, or are they introduced by the authors, so they  themselves quickly can compare statistical relevance? 

I now get the link to lower case / upper case letters from the Table caption: … lowercase letters indicate the statistical significance within treatments, uppercase letters indicate statistical significance within parts of the plant

But with which numerical value are the letter a,b,c and A,B,C associated with? The caption states … different letters indicate significantly different values p<0.005 …, which to me doesn’t make much sense, since it is only a single value but there are three different entry types (letters).

And I am completely at a loss what the double entries in Table S3(a) actually mean: for example, (i) for the “root”-entries for Ni, these are aA & B; (ii) for some other elements different letters appear; and (iii) at times only one or two of the treatments show such a double entry. What on earth does that all mean? Nowhere is an explanation given. I mentioned this already in my first review, but still no explanation has been given

Answer 8

After performing the two-way ANOVA that relates the two independent variables treatments and plant parts, we performed for each independent variable the post hoc Tukey HSD test obtaining thus two sets of grouping: one for the independent variable “plant parts” to which we decided to give the lower-case letters a, b, c, and one for the independent variable “treatments” to which we decided to give the upper-case letters A, B, C. Of course, there could be multiple entries for both the sets of letters if a number for a certain independent variable has a value in between two other values.

However now all this is useless because as the Reviewer 1 suggested the Authors transformed table S3a,b in a set of graphs, so now maybe the meaning of the letters on the bards of the graph will be easier to understand.

Question 9

Finally, having struggled significantly to plough through the vast number of tabulated numerical values and trying to link them to the narrative in the text, I think the presentation should be made more transparent. My advice would be: display the data in bar-graph form, as was done for the molecular species in Figures 3 & 4. That would have made it much easier, at least for me, to visualise the results in a comparative manner – extracting trends from a maze of numbers is extremely more difficult, and thus blurs understanding, than from a concise graphical display.

Answer 9

The authors decided to agree with this suggestion and now table S3a, b has become Figure S4 a, b Chemical analyses of the elements in the various organs according to treatment and FigureS5 a, b. Chemical analyses of the elements in the various organs according to plant parts. Different letters are for significantly different values with p<0.005 according to ANOVA and Tukey SHD post hoc test. The asterisks near the elements indicate those that have an interaction between treatment and part of the plant.

Question 10 

Figure S1 – briefly mentioned in [line 124]

Figures S2/S3 – addressed in short sentences associated with their referencing in [lines 134, 483]

These tables are OK-ish to me. However, personally I think that a bit more explanation is required; the guesses are rather hand-waving, and given the wealth of measurements / data the authors have undertaken / accumulated, more could have been extracted (I can’t easily make out any major conclusion with respect to these figures in the discussion section). Also, the sentence preceding the referencing in line 483 is garbled: it simply doesn’t make sense to me.

Answer 10

 We have already added a sentence on figure S1 in lines 483-485 of the Discussion.

 We have eliminated the garbled sentence and substituted it with the following discussion on Figure S2 and S3: “In Figure S2, where the data are grouped per plant parts the set of data are more compact indicating less difference among the plant parts within the different treatments. In Figure S3, where the data are grouped according to the treatment the data are more scattered indicating more difference in the plant parts when grouped according to the treatment.”

Question 11 

Figure S4 – addressed in [lines 484-486]

Neither from the three lines of text, nor from the 3D display of PCA components in the figure, do I understand what one may learn from the analysis data and its display.

To me the elemental and physiological (molecular) data vectors look like a seemingly random scatter of end-points. There is no mention which actual data found their way into this analysis, and no explanation of any potential significance of the data distribution, other than that the clouds” seem to be at slightly different locations in the 3D plots.

What is the point? Either substantial explanation / discussion needs to be provided, or any link between certain parts in the discussion in the manuscript text be established (I can’t see any); or the figures should be eliminated – to simply show them because application of PCA was possible is futile.

Answer 11

We have eliminated the PCA Figures from the supplementary and all their mentioning in the text.

Reviewer 3 Report

line 41-42 , combine with previous paragrapgh.

resubmit the manuscript with changes in red color, How I identified the changes?

Author Response

Answers to reviewer

Reviwer3

Question 1

line 41-42, combine with previous paragraph.

Answer 1

We have united the lines as suggested. See text.

Question 2

Resubmit the manuscript with changes in red color, How I identified the changes?

Answer 2

We have made a pdf version of the file with the track changes in red attached to this file.
